# Orochi: Versatile Biomedical Image Processor

**Gaole Dai**[1,*]    **Chenghao Zhou**[1,2,*]    **Yu Zhou**[3,*]

**Rongyu Zhang**[1]    **Yuan Zhang**[1]    **Chengkai Hou**[1]    **Tiejun Huang**[1]

**Jianxu Chen**[3,✉]
jianxu.chen@isas.de

**Shanghang Zhang**[1,✉]
shanghang@pku.edu.cn

[1] State Key Laboratory of Multimedia Information Processing,
School of Computer Science, Peking University
[2] Academy for Advanced Interdisciplinary Studies, Peking University
[3] Leibniz-Institut für Analytische Wissenschaften – ISAS – e.V.

## Abstract

Deep learning has emerged as a pivotal tool for accelerating research in the life sciences, with the low-level processing of biomedical images (e.g., registration, fusion, restoration, super-resolution) being one of its most critical applications. Platforms such as ImageJ (Fiji) and napari have enabled the development of customized plugins for various models. However, these plugins are typically based on models that are limited to specific tasks and datasets, making them less practical for biologists. To address this challenge, we introduce **Orochi**, the first application-oriented, efficient, and versatile image processor designed to overcome these limitations. Orochi is pre-trained on patches/volumes extracted from the raw data of over 100 publicly available studies using our Random Multi-scale Sampling strategy. We further propose Task-related Joint-embedding Pre-Training (TJP), which employs biomedical task-related degradation for self-supervision rather than relying on Masked Image Modelling (MIM), which performs poorly in downstream tasks such as registration. To ensure computational efficiency, we leverage Mamba's linear computational complexity and construct Multi-head Hierarchy Mamba. Additionally, we provide a three-tier fine-tuning framework (Full, Normal, and Light) and demonstrate that Orochi achieves comparable or superior performance to current state-of-the-art specialist models, even with lightweight parameter-efficient options. We hope that our study contributes to the development of an all-in-one workflow, thereby relieving biologists from the overwhelming task of selecting among numerous models. Our pre-trained weights and code will be released.

## 1 Introduction

With the rapid advancement of deep learning, modern neural networks have demonstrated remarkable scalability and spawned a wide array of downstream applications in AI for Life Science [1, 2, 3]. Among these, biomedical image processing is a pivotal topic. Its significance arises from the inherent

---

[*]Equal Contribution, ✉Corresponding Author

39th Conference on Neural Information Processing Systems (NeurIPS 2025).

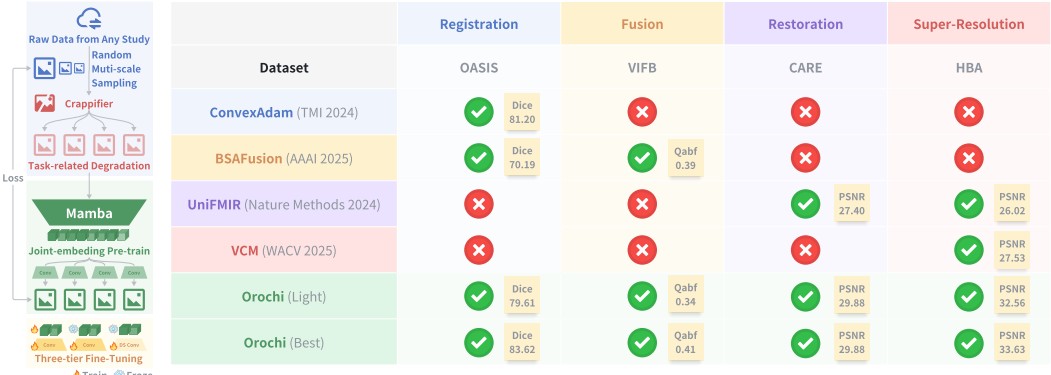

Figure 1: **Trend of Versatile Biomedical Image Precessor.** We listed the recent advancements in biomedical image processing, where matched row-to-column colour coding highlights the main task of each model. Stickers display the reported scores from the respective papers. Orochi extends the versatile bandwidth and exhibits exceptional performance across tasks and tuning modes.

constraints in acquiring biomedical images compared to natural images, which often compromise source image quality. Specifically, the most common limitations stem from imaging device operational trade-offs. For instance, in optical microscopy, excessive laser intensity can damage target tissues, while insufficient laser power introduces low signal-to-noise ratios [4]. Similarly, in computed tomography (CT), thinner slice scans subject patients to prolonged high radiation exposure, posing health risks, whereas sparse slicing results in low-resolution data [5]. These challenges drive the demand for biomedical image **restoration** [6, 4, 7, 8, 9] and **super-resolution** [10, 11, 12, 13, 5] tasks. Another class of limitations originates from the intrinsic shortcomings of imaging modalities. For example, CT imaging is efficient and provides clear hierarchical information but suffers from poor soft-tissue contrast, in contrast, magnetic resonance imaging (MRI) excels in soft-tissue resolution but requires longer acquisition times and is susceptible to motion artifacts. Such modality-specific weaknesses necessitate biomedical image **fusion** tasks [14, 15, 16, 17, 18, 19, 20, 21, 22, 23, 24, 25]. Furthermore, acquiring synchronous multi-modal data imposes demands on equipment and environments, making asynchronous data more prevalent. However, misalignment exists between tissues or cells due to temporal/specimen variability, motivating image **registration** [26, 27, 28, 29, 30, 31, 32, 33] tasks to align asynchronous or even heterogeneous datasets of the same specimen.

With the emergence of long-range dependency models [34, 35, 36] and self-supervised pre-training methods [37, 38, 39, 40], models designed for the aforementioned issues have advanced rapidly, giving rise to powerful specialist models (see Figure 1). However, we argue that in practical applications, these specialist models neglect three critical factors: **(1) Task Perspective:** Real-world biomedical imaging tasks often require multiple sequential steps (e.g., registration followed by fusion, as discussed earlier). **(2) Degradation Perspective:** Since the underlying causes of degradation share similarities, these degradations are interrelated—for example, both low signal-to-noise ratio and low resolution result in information loss. **(3) Data Perspective:** Due to their characteristics of being multi-channel, large-scale, and high-throughput, biomedical images are considerably larger than natural images, making the training and inference of multiple specialist models highly inefficient. From both efficiency and effectiveness standpoints, these issues collectively motivate the development of a universal foundational model. We aim for such a generalist model to optimize the aforementioned challenges by: **(1)** handling diverse low-level tasks within a unified framework, thereby avoiding the difficulties of selecting and integrating several specialist models; **(2)** capturing more generalized and robust features via cross-task learning during the pre-training phase; and **(3)** addresses real-world biomedical data processing costs to reduce redundant training and inference.

Therefore, we introduce **Orochi** (named after the legendary multi-headed serpent). To fulfill the envisioned goals, our design emphasizes four aspects (see Figure 2): **(1) Dataset Level:** We extensively employ unlabeled raw data from over 100 publicly available studies (see Appendix A.2) and perform our Random Multi-scale Sampling, which considers the different scales of Region-of-Interest (ROI). **(2) Pre-training Level:** Inspired by Joint-embedding Prediction Architecture (JEPA) [40], where different degradations serve as context for each others. Our Task-related Joint-

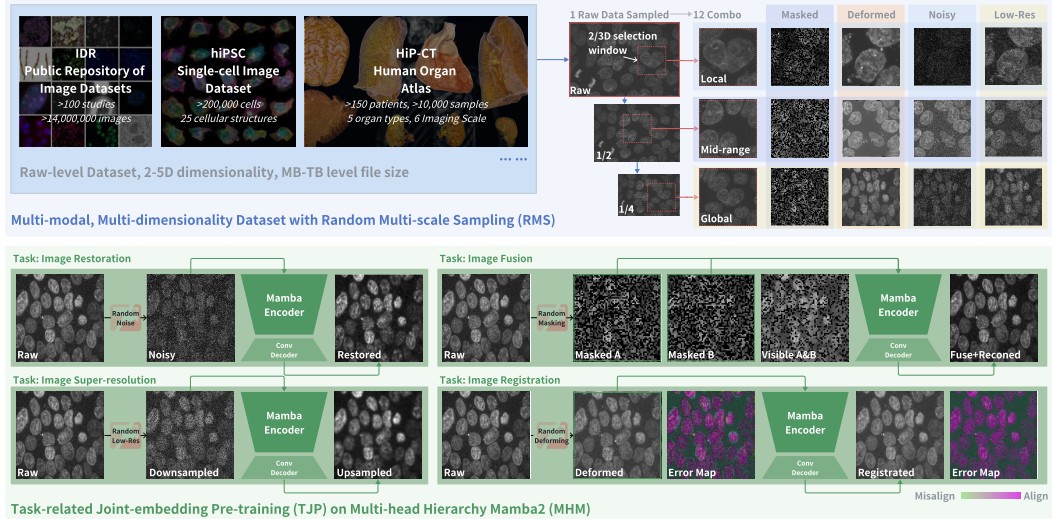

Figure 2: **Overview of the Construction of Orochi.** The upper panel illustrates the data conversion pipeline, taking into account patches/volumes at multiple scales. The lower panel presents the self-supervised strategies utilized during pre-training. Additionally, we provide supplementary images (e.g., Visible A+B, Error Map) to facilitate the comparison between inputs and outputs.

embedding Pre-training (TJP) applies various forms of task-specific degradation, and the model learns from reconstructing them jointly. **(3) Model Level:** On one hand, we employ Mamba [41] as the building blocks to leverages its linear complexity [42, 41, 43, 44]. On the other hand, the overall structure draws inspiration from the hierarchical design of the Swin-Transformer [36] by incorporating patch merging to enhance model efficiency further. **(4) Post-training Level:** We propose a three-tier fine-tuning framework to reduce the tuning cost. Ranging from full fine-tuning (Full), to fine-tuning only the replaced dense convolution head (Normal), and finally to the most lightweight variant using depth-wise separable convolution [45] (Light), thereby achieving Parameter-Efficient Fine-Tuning (PEFT) [46]. To this end, we hope that Orochi will distinguish itself as an exceptional tool among the extensive array of plugins available on platforms such as ImageJ (Fiji) [47] and napari [48], further advancing towards a user-friendly workflow with unified functionalities.

In summary, our main contributions are as follows:

1. We systematically review the significance of low-level biomedical image processing and highlight that even in this era of powerful foundational models, the paradigm centred on specialist models still exhibits inherent deficiencies. Limiting both effectiveness and efficiency from the perspectives of task, degradation, and data. **To the best of our knowledge, Orochi is the first versatile foundational model addressing these issues.**

2. We curated raw-level datasets from over 100 studies [49, 50, 51], covering a wide range of imaging modalities from 2-5D — with a total data size over 100 terabytes. During training, we introduce Random Multi-scale Sampling to achieve a unitedly raw data conversion into training patches/volumes. These converted data are used for both local and stream training, alleviating the challenges with the transmission and storage of extremely large datasets.

3. We propose Task-related Joint-embedding Pre-training (TJP), which directly learns the inter-relations among various task-specific degradations rather than relying on common Masked Image Modelling (MIM). For the model architecture, we leverage the linear complexity of Mamba and design a multi-head hierarchical structure to minimize the costs of training and inference. Finally, for post-training, we introduce a three-tier fine-tuning framework and demonstrate that even the most lightweight depth-separable convolution tuning can achieve performance comparable to existing state-of-the-art specialist models.

## 2 Related Works

**Self-supervised Learning**   Self-supervised Learning (SSL) extracts inherent data properties. Masked Image Modelling (MIM) predicts masked image regions from original pixel values using an encoder-decoder architecture, with loss in image space [37, 38]. Contrastive Learning (CL) aligns representations of augmented views of the same image in an embedding space via specialized objectives [52, 39]. Combining these, the Joint-Embedding Predictive Architecture (JEPA) [40] predicts full latent representations from context to learn robust image representations.

**Restoration**   To address low image quality in fluorescence microscopy, Content-aware image restoration (CARE) [4] uses CNNs. Li *et al.* [8] improved axial resolution using a CARE-based model with physically acquired ground truth. Subsequent works integrated Swin-Transformers [36] for efficiency (SwinIR [9]) or Mamba blocks [42] for long-range dependency modeling (MambaIR [7]). UniFMIR [6] demonstrated that pre-trained foundation models generalize well for this task.

**Super-resolution**   Super-resolution aims to overcome optical limits. DeepLP [53] employs point-scanning for reconstruction. Diffusion-based models, including volumetric conditioning modules [5] and latent diffusion in InverseSR [11], show promise for 3D brain MRI. Other approaches include local implicit image functions for flexible resolution enhancement [13], joint super-resolution and synthesis frameworks for isotropic volumes [12], and methods for multimodal image super-resolution [10].

**Registration**   Image registration aligns images by optimizing a deformation field. VoxelMorph [54] provides a learning-based 3D framework. Dual-encoder U-Nets [26], Swin-Transformers for long-distance correspondences (TransMorph [32]), and Mamba blocks [42] for efficient long-range modeling (MambaMorph [33]). Fast 3D registration methods have been proposed by Siebert *et al.* [27, 29], while Mok *et al.* [30, 28] address large deformations with Laplacian Pyramid Networks.

**Fusion**   Multi-modality image fusion integrates complementary information. Techniques include bidirectional stepwise feature alignment for unaligned images (BSAFusion [25]), mutual enhancement for PAT/MRI fusion [24], and diffusion-based methods incorporating fusion priors (Diff-IF [55]) or denoising diffusion models [16]. Semantic-aware strategies with registration are found in SuperFusion [21] and MURF [22]. Other notable methods encompass one-stage progressive dense registration [23], U2Fusion [14], and Equivariant fusion [15]. Diverse strategies also include lightweight and semantic-guided approaches (ALMFNET [17], MSGFUSION [18]), dictionary-based and GAN-driven frameworks [19, 56], and unsupervised methods [20].

## 3 Methods

Due to page limitations, this section primarily emphasizes our comprehensive degradation designs used for self-supervision. The Appendix provided detailed architecture of the Multi-head Hierarchy Mamba model along with the three-tier fine-tuning framework B.

### 3.1 Preliminary: Self-supervised Degradation

Self-supervised image learning can be generally formulated as learning a reconstruction function $f_\theta$ that recovers the original image $x$ from its degraded $D(x)$. Formally, this objective is defined as:

$$\min_\theta \ \mathbb{E}_x \left[ \ell \Big( x, f_\theta \big( D(x) \big) \Big) \right], \tag{1}$$

where $x$ is the sampled data, $p_{\text{data}}$, $D(\cdot)$ denotes a degradation function applied to $x$, $f_\theta$ is the parameterized model, and $\ell$ is a loss function (e.g., the L2 loss or perceptual loss).

**Masked Image**   For masked image degradation, the degradation function is defined as: $D_{\text{mask}}(x) = x \odot M$, where $M \in \{0, 1\}^{H \times W}$ is a binary mask with height $H$ and width $W$ that selectively occludes regions of $x$. This degradation helps the model learn to infer missing information.

**Deformed Image**   For deformed image degradation, the degradation function takes the form: $D_{\text{def}}(x) = \mathbf{T}(x)$, where $\mathbf{T}(\cdot)$ represents a spatial transformation (such as rotation, scaling, or warping). This degradation introduces geometric distortions that mimic real-world variations.

**Nosiy Image** For noisy image degradation, the degradation function is defined as: $D_{\text{noise}}(x) = x + \eta$ where $\eta$ denotes additive noise (typically Gaussian noise), simulating sensor imperfections or environmental interference.

**Low-resolution Image** For low-resolution image degradation, the degradation function is given by: $D_{\text{LR}}(x) = \downarrow_s (x)$, where $\downarrow_s$ is a down-sampling operator with scale factor $s$, reducing the resolution of $x$ to simulate the effects of low-resolution imaging.

### 3.2 Orochi: Random Multi-scale Sampling

Random Multi-scale Sampling aims to extract patches/volumes with diverse scales from raw images. Given a raw image $I$, the procedure consists of two main steps: **(1) Multi-scale Resizing:** We first generate scaled versions of the raw image $I$ to capture features at different resolutions. In particular, we resize $I$ to scales $1/2$ and $1/4$ of its original size. Formally, let:

$$I_s = \downarrow_s (I), \quad s \in \{1, \tfrac{1}{2}, \tfrac{1}{4}\}, \tag{2}$$

where $\downarrow_s (\cdot)$ denotes down-sampling with factor $s$. **(2) Random Window Sampling:** For each scaled image $I_s$, we define a fixed-size window $K$ (compatible with the pre-training requirements in either 2D or 3D) and perform random sampling to extract sub-patches. Let the window $K$ have dimensions $W \times H$ (or $W \times H \times D$ for 3D data). A randomly sampled 2D patch $x_s$ at scale $s$ is given by:

$$x_s = I_s\big(i : i + W - 1, \; j : j + H - 1\big), \tag{3}$$

where $(i, j)$ is a randomly chosen starting coordinate in $I_s$.

Collectively, the set of patches extracted across scales is represented as:

$$x = \{x_{s,n} \mid s \in \{1, \tfrac{1}{2}, \tfrac{1}{4}\}, \; n = 1, \ldots, \text{N}_s\}, \tag{4}$$

where $\text{N}_s$ denotes the number of patches sampled from the image at scale $s$. These multi-scale patches are then passed to subsequent degradation processes (e.g., masking, deformation, noise addition, and low-resolution conversion). By performing random sampling across multiple scales, our method extended the data diversity and enabled more robust feature learning across various datasets.

### 3.3 Orochi: Task-related Joint-embedding Pre-training

**Dual-Masking Reconstructive Fusion** To better address the biomedical image fusion task, where the combination of existing contexts is crucial, we modified the conventional Masked Image Modelling approaches [37, 38], which typically employ a single masking strategy. Specifically, we applied two distinct masking operations to the training data $x$, thereby generating two independent masks:

$$x_A = x \odot M_A, \quad x_B = x \odot M_B, \tag{5}$$

where $M_A, M_B \in \{0, 1\}^{H \times W}$ are binary masks with only partial overlap and ensure invisible information retention even after fusion. The masking probabilities are generated by:

$$M_k[i, j] = \mathbf{1}[\xi_{i,j}^k < \tau], \quad k \in A, B, \tag{6}$$

where $\xi_{i,j}^k \sim \mathbf{U}(0, 1)$ represents a random value extracted from a uniform distribution for grid coordinates $i, j$, and $\tau$ is the masking threshold. The key innovation is that our model is exposed to process both masked inputs $(x_A, x_B)$ simultaneously to recover the original image: $\hat{x} = f_\theta(x_A, x_B)$. This guides the model to develop robust feature extraction capabilities that can identify complementary information across different masked views, and then fuse these partial observations coherently to reconstruct missing regions in both inputs.

**Spatially-varying Gaussian down-sample** For down-sampling, we adapt similar principles from DeepLP [53], which tested noisy down-sampling beyond uniform down-sampling in self-supervised microscopy restoration. We enhance this noisy down-sampling with spatially varying characteristics:

$$D_{\text{LR}}(x) = \mathbf{G}\sigma\text{var}(\uparrow_{\frac{1}{s}} (\downarrow_s (x + \eta))), \tag{7}$$

where $\downarrow_s$ represents down-sampling with a random scale factor $s$, $\uparrow_{\frac{1}{s}}$ denotes upsampling back to the original resolution, $\eta \sim \mathbf{N}(0, \sigma_{\text{down}}^2)$ is normal distributed noise added during the down-sampling process with $\sigma_{\text{down}} \sim \mathbf{U}(0.01, 0.1)$, $\mathbf{U}$ represent uniform distribution, and $\mathbf{G}\sigma\text{var}$ denotes spatially-varying Gaussian filtering. It can be defined as:

$$\mathbf{G}\sigma\text{var}(x)[i,j] = \sum_{u,v} g_{\sigma(i,j)}(u,v) \cdot x[i-u, j-v], \tag{8}$$

where $g_\sigma$ represents a Gaussian kernel (2/3D) with standard deviation $\sigma(i,j) \sim \mathbf{U}(\sigma_{\min}, \sigma_{\max})$ that varies across grid coordinates $i, j$. This mimics the heterogeneous blurring found in optical systems.

**Multi-scale Smoothed Perlin Noise Deformation**   For the self-supervised registration task, constructing a realistic deformation field is important. We conducted multi-scale Perlin noise fields that simulate the hierarchy variations in natural anatomical structures. Given an image $x$, we generate a deformation field $\Phi$ and its corresponding deformed image $D_{\text{def}}(x)$ as follows:

$$D_{\text{def}}(x) = \mathbf{T}(x, \Phi), \quad \Phi = \mathbf{G}_\sigma(\mathbf{Per}(\mathbf{f}, \mathbf{p})), \tag{9}$$

$\mathbf{T}(\cdot, \cdot)$ is a spatial transformation operator, $\mathbf{G}_\sigma(\cdot)$ denotes spatially-varying Gaussian smoothing with parameter $\sigma$, and $\mathbf{Per}(\mathbf{f}, \mathbf{p})$ represents multi-octave Perlin noise with frequency $\mathbf{f}$ and persistence $\mathbf{p}$.

The multi-octave Perlin noise is specifically defined as:

$$\mathbf{Per}(\mathbf{f}, \mathbf{p}) = \sum_{n=1}^{\mathrm{N}} \mathbf{p}^{n-1} \cdot \mathbf{S}(\mathbf{f}^{n-1} \cdot (i,j)), \tag{10}$$

where $\mathbf{S}(\cdot)$ is the simplex noise function, N is the number of octaves and **coords** represents the grid coordinates. This multi-scale approach generates deformation fields with varying levels of detail.

To enhance the anatomical plausibility of the deformations, we apply normalization and bound it using a tanh function: $\Phi_{\text{final}} = \alpha \cdot \tanh(\Phi)$, where $\alpha$ controls the maximum displacement magnitude.

**Multi-stage Noise Simulation**   To simulate realistic noise, we adopted a multi-stage process:

$$D_{\text{noise}}(x) = \mathbf{Bi}_p(\mathbf{Poi}(\max(0, x + \eta))), \tag{11}$$

where $\eta \sim \mathbf{N}(0, \sigma_{\text{noise}}^2)$ with $\sigma_{\text{noise}} \sim \mathbf{U}(0.075, 0.15)$ represents Gaussian noise, $\mathbf{Poi}(\lambda)$ denotes Poisson noise with intensity parameter $\lambda$ (modeling photon-counting statistics), and $\mathbf{Bi}_p$ represents binary (salt-and-pepper) noise that affects a proportion of pixels with probability $p$.

These sophisticated degradation designs enable our framework to simulate a wide spectrum of real-world imaging artifacts, encouraging the model to handle diverse image quality issues encountered.

## 4   Experiments

We conducted comprehensive comparisons strictly following the setups in published specialist models (UniFMIR [6], VCM [5], Transmorph [32], and BSAFusion [25], see Appendix A.4 for details). Resulting in more than 30 state-of-the-art baselines across multiple benchmarks for various biomedical image-processing tasks to demonstrate the effectiveness and versatility of Orochi. We color-coded the performance in Table 1, 2, 3, and 4 with **Red (1st)**, Blue (2nd), and the row color reflects the training type with [Training-free], [Training from Scratch], [Fine-Tuning], and [Efficent Fine-Tuning]. See the Appendix for more details of the experiment setups A and extra validation C.

**Generalization Capability on In-Domain Data**   Given that our model, Orochi, is extensively pre-trained, we expect it to exhibit strong generalization capabilities on in-domain data. Accordingly, in Figure 3 we demonstrate Orochi's zero-shot performance on various stained microscopy images [51] (results on clinical images [50] are detailed in the Appendix C.1). Panels (A)–(D) illustrate Orochi's robust processing capabilities. In Panel (E), we further examine whether these outcomes align with our algorithmic expectations. For example, our Dual-Masking Reconstructive Fusion anticipates that the model learns an effective fusion strategy and leverages the existing information from both

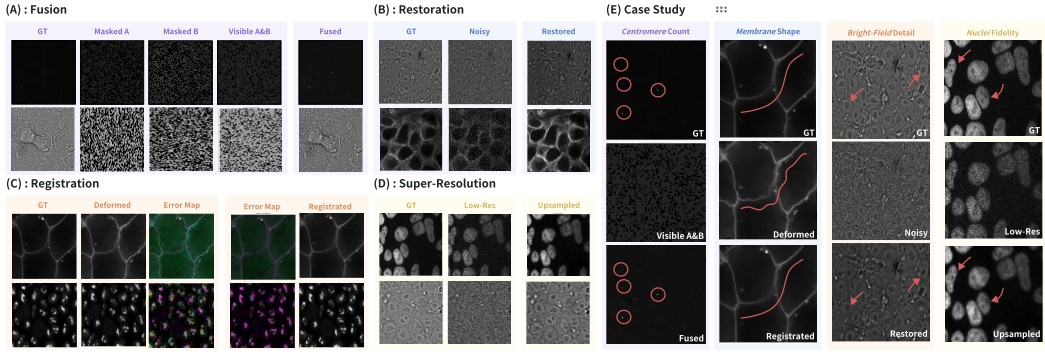

Figure 3: **In-Domain Generalization Performance of Orochi on Unseen Test Images.** (A)–(D) illustrate Orochi's robust performance across various low-level processing tasks when applied to unseen testing images after pre-training. Supplementary images include the dual-masking images and naive merge results for the fusion task. Error maps for the registration task. (E) provides in-depth case studies: for the fusion task, the centromere count is emphasized in both the reconstructed image and the original image (highlighted with circles); for the registration task, subtle deformations of the cell membrane are accentuated; and for the restoration and super-resolution tasks, the fine details of bright-field images and the internal structures of DNA-stained cell nuclei are emphasized

Table 1: **Isotropic 3D volume Restoration Task.** Low-high laser data pairs along the XY axis are collected and serve as the training set. However, the evaluation is on both XY slices and XZ slices.

| Method | PSNR (XY) ↑ | SSIM (XY) ↑ | PSNR (XZ) ↑ | SSIM (XZ) ↑ |
|---|---|---|---|---|
| Dataset: *CARE* [4] | | | | |
| Li *et al.* [8] | 23.71 | 0.58 | 24.51 | 0.58 |
| CARE [4] | 25.60 | 0.60 | 25.76 | 0.64 |
| SwinIR [9] | 25.98 | 0.62 | 26.41 | 0.64 |
| MambaIR [7] | 25.89 | 0.64 | 27.17 | 0.65 |
| UniFMIR [6] | 27.12 | 0.66 | 27.67 | 0.66 |
| prune-UniFMIR (FP16) [6] | 25.00 | 0.59 | 26.48 | 0.64 |
| prune-UniFMIR (FP32) [6] | 26.18 | 0.63 | 26.24 | 0.64 |
| **Orochi** (Full) | 28.31 | 0.70 | 28.52 | 0.71 |
| **Orochi** (Normal) | 29.15 | **0.71** | 29.43 | 0.71 |
| **Orochi** (Light) | **29.77** | **0.71** | **29.98** | **0.72** |

Table 2: **MRI Axial Super-resolution Task.** Two intensities of low-resolution data are trained and tested in this task, with 4mm (i.e x4 down-sampled) and 8mm (i.e x8 down-sampled)

| Method | PSNR (4mm) ↑ | SSIM (4mm) ↑ | PSNR (8mm) ↑ | SSIM (8mm) ↑ |
|---|---|---|---|---|
| Dataset: *HBA* [57] | | | | |
| Cubic | 23.84 | 0.76 | 21.80 | 0.63 |
| UniRes [10] | 21.49 | 0.69 | 20.91 | 0.63 |
| SynthSR [12] | 19.22 | 0.66 | 19.02 | 0.62 |
| LIIF [13] | 32.41 | 0.95 | 25.12 | 0.81 |
| InverseSR-LDM [11] | 28.59 | 0.80 | 27.92 | 0.75 |
| InverseSR [11] | 27.51 | 0.88 | 23.66 | 0.79 |
| VCM [5] | 27.54 | 0.87 | 27.52 | 0.86 |
| **Orochi** (Full) | **35.33** | 0.95 | **31.93** | 0.89 |
| **Orochi** (Normal) | 34.60 | **0.96** | 29.51 | 0.89 |
| **Orochi** (Light) | 34.83 | **0.96** | 30.28 | **0.90** |

sources before reconstruction, rather than reconstructing masked regions separately. This expectation is validated in the *Centromere* count case study, where masked A and B each exhibit a partial absence of centromeres, and the **model successfully performs complementary fusion and the final reconstruction does not arbitrarily generate *Centromeres* across extensive background regions.** This precise control over fine structural details is also evident in other cases.

**Image Restoration Task**    In Table 1, we present the performance of Orochi on the isotropic 3D volume restoration task. In microscopy imaging, the image quality along the XY plane is typically much higher than that along the XZ plane due to the inherent limitations of sequential (layer-by-layer) imaging, such as in light-sheet microscopy, which leads to the formation of isotropic data. To address this, CARE [4] leverages the high-resolution XY data for training and subsequently restores the lower-resolution XZ data. On this task, Orochi not only significantly outperforms **train-from-scratch** models like SwinIR [36] (+2.11 PSNR) and MambaIR [7] (+1.35 PSNR), but it also comprehensively surpasses **pre-trained foundation model** UniFMIR [6] across both fully fine-tuned (+0.85 PSNR) and efficiently fine-tuned (+3.19 PSNR) configurations. An intriguing finding is that our results indicate Orochi with PEFT leads the list. This outcome is plausible given that the dataset, derived from

Table 3: **Inter-patient Brain Registration Task.** During training, the model goal is to input paired MRI data from distinct patients and output prediction of the registration flow. This flow is applied to the corresponding segmentation data to calculate the dice loss. Thereby, regional deformation can be learned with supervision.

| Method | Dice ↑ | HD95 ↓ | SDlogJ ↓ |
|---|---|---|---|
| **Dataset: *OASIS* [58]** | | | |
| Initial | 56.10 | 3.86 | — |
| Lv *et al.* [26] | 80.00 | 1.77 | 0.08 |
| Siebert *et al.* [27] | 81.00 | 1.63 | 0.07 |
| Mok *et al.* [28] | 82.00 | 1.67 | 0.07 |
| PIMed [31] | 78.76 | 1.86 | **0.06** |
| LapIRN [30] | 82.18 | 1.67 | 0.08 |
| ConvexAdam [29] | 81.20 | 1.71 | 0.07 |
| Transmorph-B [32] | 81.62 | 1.69 | 0.12 |
| Transmorph-L [32] | 82.22 | 1.66 | 0.12 |
| Mambamorph [33] | 81.81 | 1.66 | 0.09 |
| **Orochi** (Full) | **83.62** | **1.60** | 0.11 |
| **Orochi** (Normal) | 82.52 | 1.65 | 0.12 |
| **Orochi** (Light) | 79.61 | 1.73 | **0.06** |

Table 4: **CT-MRI Fusion Task.** Volumetric MRI and CT data are sent jointly to the model, reconstructing a single fused result. This fused result would be compared with both MRI and CT input for similarity calculation (e.g. SSIM).

| Method | $Q_{abf}$ ↑ | $Q_{cv}$ ↓ | SSIM ↑ |
|---|---|---|---|
| **Dataset: *VIFB* [57]** | | | |
| U2Fusion [14] | 0.32 | 6,580.80 | 0.41 |
| EMMA [15] | 0.29 | 6,695.80 | 1.18 |
| ALMFnet [17] | 0.29 | 7,200.50 | 1.27 |
| MsgFusion [18] | 0.19 | 7,090.40 | 0.29 |
| MDHU [19] | 0.22 | 7,417.60 | 1.23 |
| UMF-CMGR [20] | 0.25 | 4,638.70 | 1.35 |
| SuperFusion [21] | 0.28 | 4,828.90 | 0.97 |
| MURF [22] | 0.33 | 5,554.60 | 1.27 |
| IMF [23] | 0.27 | 4,439.60 | 1.34 |
| PAMRFuse [24] | 0.09 | 5,408.00 | 0.18 |
| BSAFusion [25] | 0.39 | 4,155.10 | 1.38 |
| DDFM [16] | 0.26 | 5,981.40 | 1.31 |
| **Orochi** (Full) | **0.41** | **2,351.57** | 1.39 |
| **Orochi** (Normal) | 0.37 | 2,519.36 | **1.45** |
| **Orochi** (Light) | 0.34 | 2,461.41 | 1.43 |

isotropic data pairs, comprises fewer than 100 total training patches. Consequently, Full Fine-Tuning or training from scratch is prone to over-fitting. (see Appendix C.3 for extra comparisons)

**Image Super-resolution Task**    We next evaluated the image super-resolution capabilities of Orochi (see Table 2). Early super-resolution models typically rely on CNN-based architectures such as UniRes [10] and SynthSR [12], which are efficient yet often lack sufficient expressiveness and generalization ability. LIIF [13] leverages the power of Implicit Neural Representations (INR) to perform implicit interpolation; however, the high training cost associated with INR limits its adaptability to real-world scenarios. More recent approaches, including InverseSR [11] and VCM [5], based on powerful pre-trained Brain-Latent Diffusion Models (LDM) [59] to overcome these shortcomings. In this setting, Orochi significantly outperforms all the aforementioned architectures. At an 8mm slice thickness, Orochi achieves a PSNR that is 4.01 points higher than InverseSR and 2.76 points higher than VCM. These gains demonstrate that **among pre-trained models, Orochi's pre-training is markedly superior to that of Brain-LDM, both in terms of the pre-training data and purpose.**

**Image Registration Task**    We further evaluated the registration task using the dataset from Learn2Reg [60] (see Table 3). In this task, brain MRI images from different patients (i.e., inter-patients) are registered (see Appendix C.2 for patient-to-atlas brain registration test), and the model's ability to handle subtle deformations is assessed by measuring the similarity of the segmented brain regions after registration (e.g. Dice). Biomedical image registration has evolved from CNN-based [29, 30, 27] to Transformer-based architectures [32, 54], with even linear-complexity models such as Mamba [33] emerging in recent work. In comparison to these methods, **our approach achieves Dice scores that are 2.42 points higher than ConvexAdam, 2.0 points higher than Transmorph, and 1.81 points higher than Mambamorph.**

**Image Fusion Task**    Finally, as illustrated in Table 4, we evaluated Orochi's performance on the image fusion task. Recent trends in this domain have integrated image registration as an auxiliary task to facilitate fusion, as demonstrated by methods such as BSAFusion [25], UMF-CMGR [20], MURF [22], and SuperFusion [21]. Although these models typically exhibit limited registration capabilities (see Appendix C.2), this aligns with our pursuit of developing a versatile, comprehensive model. Compared with the recent advanced model BSAFusion, Orochi outperforms on all evaluated metrics, achieving improvements of +0.02 in $Q_{abf}$, -1803.53 in $Q_{cv}$, and +0.07 in SSIM. Combined with our state-of-the-art performance on the registration task, **these results establish Orochi as the first model in this domain to achieve such performance.**

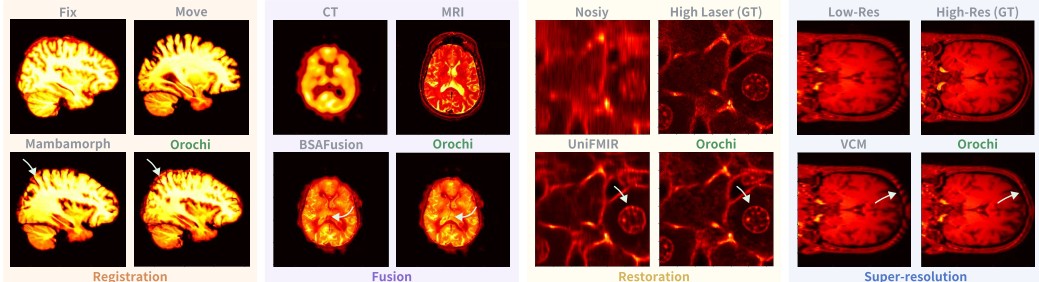

Figure 4: **Visualization Comparison to Recent Advances.** Concise visualization for each task is provided. To enhance comprehension, arrows have been incorporated to facilitate evaluation.

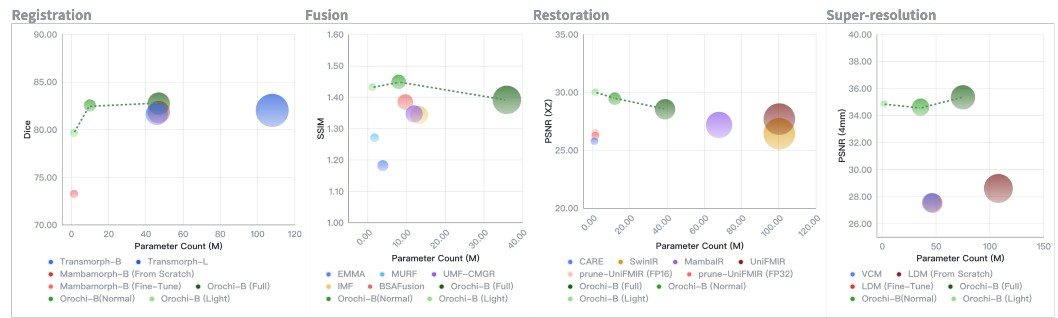

Figure 5: **Fine-Tuning Efficiency V.S Performance.** We present the training parameter efficiency of various models alongside their corresponding results. The three-tier fine-tuning results for Orochi (Full, Normal, Light) are illustrated using a gradient of green colours, from deep to light, and are connected by green dashed lines to indicate the trend. Other baselines are shown in the legend.

**Visualizations** In Figure 4, we provide qualitative results of Orochi. Specifically, Orochi demonstrates a superior capability in **handling subtle degradations**. (see Appendix C.1 C.2 for more)

Table 5: **Pre-train Strategies V.S Performance.** Datasets and setups remain the same as those experiments in the previous sections.

| Strategy | Registration (Dice ↑) | Fusion ($Q_{abf}$ ↑) | Restoration (PSNR ↑) | Super-Resolution (PSNR ↑) |
|---|---|---|---|---|
| **MAE** [37] (Single Mask) | 71.22 | 0.36 | 26.67 | 29.17 |
| **I-JEPA** [40] (Dual Mask) | 69.97 | 0.39 | 25.02 | 28.81 |
| **Orochi** (TJP) | **83.62** | **0.41** | **29.88** | **33.63** |

**Ablation Study - Comparison to Other Pre-train Strategies** In Table 5, we demonstrate the limitations of relying solely on Masked-image-Modelling (MIM), particularly in registration tasks. Additionally, we observe that the dual-masking approach employed in I-JEPA [40] underperforms compared to Orochi. We hypothesize that this is because chunk masking is more advantageous for high-level tasks rather than the low-level focus of our study.

**Ablation Study - Larger ≠ Better, Fine-Tuning Efficiency V.S Performance** As shown in Figure 5, the number of trainable parameters is not the decisive factor for downstream tasks—particularly in data-limited scenarios such as biomedical imaging. In many cases, opting for **Parameter-Efficient Fine-Tuning (using only 1–2% of the total parameter count) prevents overfitting and achieves both efficient and effective results.**

## 5 Conclusion

We introduce **Orochi**, the first versatile biomedical image processor designed for low-level tasks. **To enhance effectiveness,** we propose Random Multi-scale Sampling, which is a scalable way to

leverage raw data from a wide range of studies. The extracted data is then processed through our Task-related Joint-embedding Pre-training (TJP), where a unified and robust embedding is learned from various task-related degradations. **For efficiency**, we developed Multi-head Hierarchy Mamba and provide a three-tier fine-tuning framework (Full, Normal, and Light). These design choices ensure high efficiency during pre-training, post-tuning, and test inference. **Our experiments** demonstrate that Orochi exhibits in-domain generalization capability across multiple tasks and achieves state-of-the-art performance compared to specialist models with efficient fine-tuning (less than 5% of total parameters). This suggests that constructing a generalist image processor may lie **more in the diversity of the dataset and the pre-training strategy than in increasing the model size naively.**

## 6 Acknowledgements

G.D. was supported by the National Natural Science Foundation of China (Grant No. W2442028). S.Z. was supported by the National Science and Technology Major Project (Grant No. 2022ZD0117800). The work of ISAS was supported by the "Ministerium für Kultur und Wissenschaft des Landes Nordrhein-Westfalen" and "Der Regierende Bürgermeister von Berlin, Senatskanzlei Wissenschaft und Forschung." The work was further supported by the Bundesministerium für Forschung, Technologie und Raumfahrt, BMFTR under the funding reference 161L0272.

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

# A Experiment Setups

## A.1 Baselines

**Registration**

- **Lv et al.** [26]: Uses a dual-encoder U-Net for coarse-to-fine registration.
- **Siebert et al.** [27]: Proposes a fast 3D registration approach.
- **Mok et al.** [28]: Employs conditional deformable convolutions.
- **PIMed** [31]: From the Learn2Reg challenge.
- **LapIRN** [30]: Uses a Laplacian pyramid network for large deformations.
- **ConvexAdam** [29]: Adopts a dual-optimization strategy.
- **TransMorph** [32]: Based on a Transformer architecture for capturing long-range correspondences.
- **MambaMorph** [33]: Utilizes mamba blocks for efficient long-range dependency modeling.

**Fuison**

- **U2Fusion** [14]: Provides a unified unsupervised fusion approach.
- **EMMA** [15]: Employs equivariant learning for fusion.
- **ALMFNet** [17]: Searches for a lightweight generalized fusion network.
- **MsgFusion** [18]: Uses a semantic-guided two-branch network.
- **MDHU** [19]: Uses multi-dictionary learning with truncated Huber filtering.
- **UMF-CMGR** [20]: Adopts cross-modality generation and registration.
- **SuperFusion** [21]: Combines registration and fusion with semantic awareness.
- **MURF** [22]: Reinforces multi-modal registration and fusion mutually.
- **IMF** [23]: Improves fusion with a progressive dense registration strategy.
- **PAMRFuse** [24]: Focuses on feature alignment.
- **BSAFusion** [25]: Adopts bidirectional stepwise feature alignment.
- **DDFM** [16]: Utilizes a denoising diffusion model for fusion.

**Super-Resolution**

- **Cubic**: Bicubic interpolation as a traditional baseline.
- **UniRes** [10]: Designed for super-resolving multimodal clinical MRI.
- **SynthSR** [12]: Performs joint super-resolution and synthesis.
- **LIIF** [13]: Learns continuous image representations for implicit interpolation.
- **InverseSR** [11]: Uses a latent diffusion model for 3D brain MRI super-resolution.
- **VCM** [5]: Applies a volumetric conditioning module.

**Restoration**

- **Li et al.** [8]: Improves axial resolution.
- **CARE** [4]: Uses a content-aware network for fluorescence microscopy image restoration.
- **SwinIR** [9]: Employs a Swin-Transformer for efficient image restoration.
- **MambaIR** [7]: Utilizes mamba blocks for modeling long-range dependencies.
- **UniFMIR** [6]: Fine-tunes a pre-trained foundation model for generalizable fluorescence microscopy-based restoration (with pruned FP16/FP32 variants).

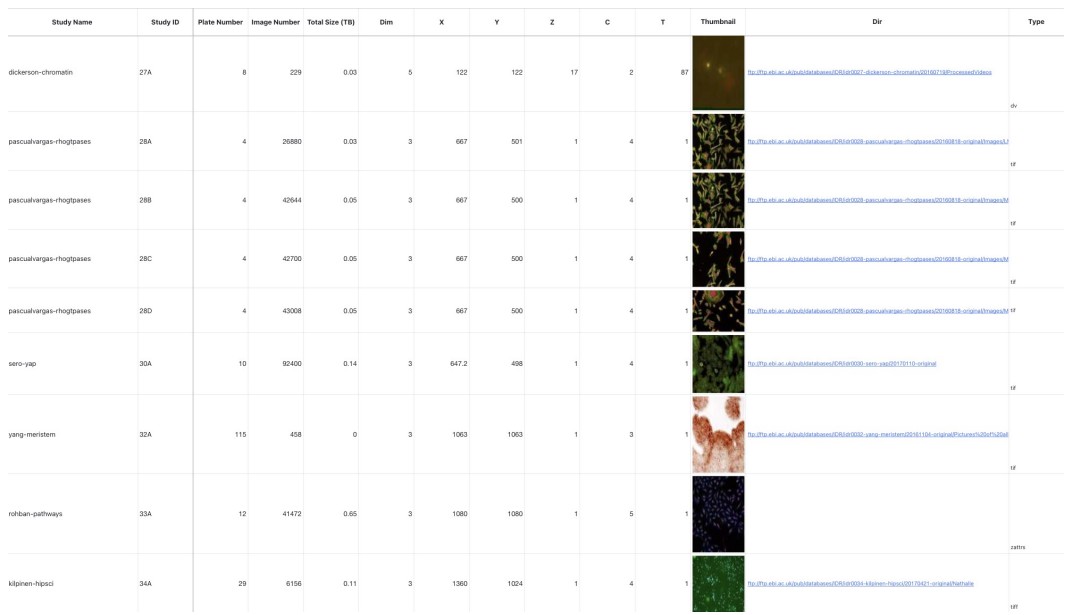

| Study Name | Study ID | Plate Number | Image Number | Total Size (TB) | Dim | X | Y | Z | C | T | Thumbnail | Dir | Type |
|---|---|---|---|---|---|---|---|---|---|---|---|---|---|
| dickerson-chromatin | 27A | 8 | 229 | 0.03 | 5 | 122 | 122 | 17 | 2 | 87 | | ftp://ftp.ebi.ac.uk/pub/databases/IDR/idr0007-dickerson-chromatin/20100719/ProcessedVideos | dv |
| pascualvargas-rhogtpases | 28A | 4 | 26880 | 0.03 | 3 | 667 | 501 | 1 | 4 | 1 | | ftp://ftp.ebi.ac.uk/pub/databases/IDR/idr0028-pascualvargas-rhogtpases/20160818-original/images/J | tif |
| pascualvargas-rhogtpases | 28B | 4 | 42644 | 0.05 | 3 | 667 | 500 | 1 | 4 | 1 | | ftp://ftp.ebi.ac.uk/pub/databases/IDR/idr0028-pascualvargas-rhogtpases/20160818-original/images/M | tif |
| pascualvargas-rhogtpases | 28C | 4 | 42700 | 0.05 | 3 | 667 | 500 | 1 | 4 | 1 | | ftp://ftp.ebi.ac.uk/pub/databases/IDR/idr0028-pascualvargas-rhogtpases/20160818-original/images/M | tif |
| pascualvargas-rhogtpases | 28D | 4 | 43008 | 0.05 | 3 | 667 | 500 | 1 | 4 | 1 | | ftp://ftp.ebi.ac.uk/pub/databases/IDR/idr0028-pascualvargas-rhogtpases/20160818-original/images/M | tif |
| sero-yap | 30A | 10 | 92400 | 0.14 | 3 | 647.2 | 498 | 1 | 4 | 1 | | ftp://ftp.ebi.ac.uk/pub/databases/IDR/idr0000-sero-yap/20170110-original | tif |
| yang-meristem | 32A | 115 | 458 | 0 | 3 | 1063 | 1063 | 1 | 3 | 1 | | ftp://ftp.ebi.ac.uk/pub/databases/IDR/idr0032-yang-meristem/20161104-original/Pictures%20of%20all | tif |
| rohban-pathways | 33A | 12 | 41472 | 0.65 | 3 | 1080 | 1080 | 1 | 5 | 1 | | | cattrs |
| kilpinen-hipsci | 34A | 29 | 6156 | 0.11 | 3 | 1360 | 1024 | 1 | 4 | 1 | | ftp://ftp.ebi.ac.uk/pub/databases/IDR/idr0034-kilpinen-hipsci/20170421-original/Nathalie | tif |

Figure 6: **Preview of the Metadata List of Studies**

## A.2   Datasets

### Pre-train

- A combined multi-modal biomedical image dataset aggregated from over 100 public studies, encompassing various imaging modalities and degradation types [51, 50, 49]. In Figure 6, we provide a preview of the metadata of the studies we covered (Excel Form would be included in the Zip file). Since our RMS method is highly scalable, we plan to further update this list in the future and explore the borderline.

### Registration

- The OASIS brain MRI dataset from the Learn2Reg 2021 challenge, used to evaluate the overlap of segmented regions and the smoothness of the deformation fields [60, 58].

### Fuison

- A CT–MRI paired fusion dataset (VIFB), which assesses the integration of complementary information across modalities [57].

### Super-Resolution

- The Harvard Whole Brain Atlas (HBA), providing high-quality MRI images for evaluating low-resolution image reconstruction [57].

### Restoration

- The CARE microscopy image dataset, used to evaluate the enhancement of low signal-to-noise ratio fluorescence microscopy images [4].

## A.3   Metrics

### Registration

- **Dice similarity coefficient:** Computed as

$$Dice = \frac{2|A \cap B|}{|A| + |B|}$$

which measures the overlap between the segmented regions.

- **95th percentile Hausdorff Distance (HD95):** Defined as the 95th percentile of the distances between boundary points of the segmented regions.

- **Standard deviation of the log-Jacobian determinant (SDlogJ):** Calculated as the standard deviation of $\log(\det(J))$, where $J$ is the Jacobian matrix of the deformation field. This metric reflects the smoothness of the deformation field [60].

**Fuison**

- $Q_{\textbf{AB/F}}$ **($Q_{abf}$):** Measures the quality of the fusion by evaluating the consistency between the fused image and the input modalities.

- $Q_{\textbf{CV}}$ **($Q_{cv}$):** Assesses the contrast consistency across the fused image.

- **Structural Similarity Index (SSIM):** Computed based on comparisons of luminance, contrast, and structure between 2 source images [25].

**Super-Resolution**

- **Peak Signal-to-Noise Ratio (PSNR):** Calculated as

$$PSNR = 10 \log_{10} \left( \frac{\text{MAX}_I^2}{MSE} \right)$$

where $\text{MAX}_I$ is the maximum possible pixel value and MSE is the mean squared error between the reconstructed and reference images.

- **SSIM:** Evaluates perceptual similarity between the super-resolved and reference images [9].

**Restoration**

- **PSNR:** As above, it measures the pixel-level fidelity between the restored image and the high-quality reference.

- **SSIM:** Measures the structural similarity between the restored and reference images [4].

## A.4 Code-base

**Pre-train**

- Adapted from public GitHub implementations of the Swin-Transformer and Transmorph. **Swin-Transformer:** `https://github.com/microsoft/Swin-Transformer` [36]; **Transmorph:** `https://github.com/junyuchen245/TransMorph_Transformer_for_Medical_Image_Registration` [32].

**Registration**

- Implemented based on the Transmorph GitHub code. **Link:** `https://github.com/junyuchen245/TransMorph_Transformer_for_Medical_Image_Registration` [32].

**Fuison**

- Built with reference to the BSAFusion GitHub code. **Link:** `https://github.com/slrl123/BSAFusion` [25].

**Super-Resolution**

- Implemented based on GitHub codes of InverseSR and VCM. **InverseSR:** `https://github.com/BioMedAI-UCSC/InverseSR` [11]; **VCM:** `https://github.com/Ahn-Ssu/VCM` [5].

**Restoration**

- Based on the UniFMIR GitHub implementation. **Link:** `https://github.com/cxm12/UNiFMIR` [6].

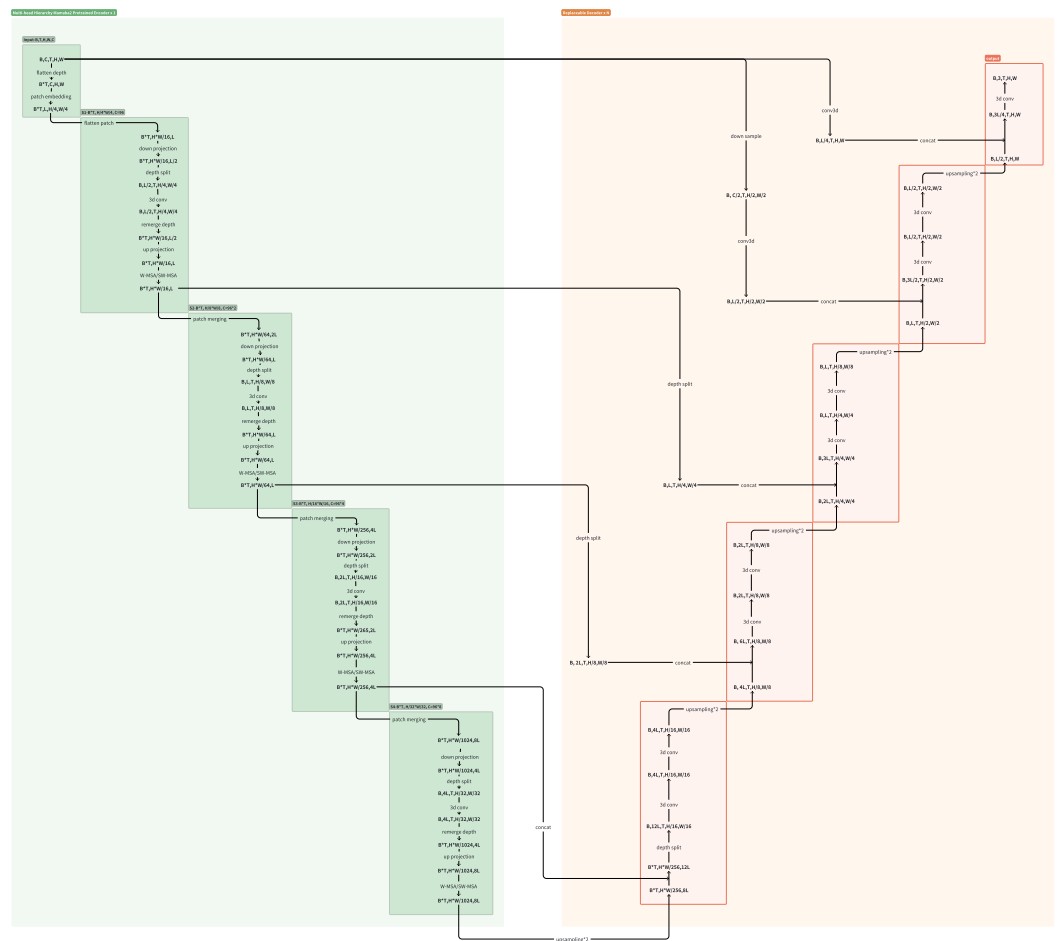

Figure 7: **Model architecture of Multi-head Hierarchy Mamba (MHM).** The model contains a unified hierarchical Mamba encoder with a replaceable decoder (e.g. regular convolution head or depth-wise separable convolution head) for tuning

## B  Experiment Configurations

**Multi-head Hierarchy Mamba & Three-Tier Fine-Tuning Framework**    Figure 7, presents a comprehensive diagram of Orochi's backbone architecture. Post-tuning, the interchangeable decoder can be replaced as required. We evaluated Orochi's performance using our Three-Tier Fine-Tuning Framework, which includes full fine-tuning (Full, 100% parameters), regular convolution head with the encoder frozen (Normal, 10-30% parameters), and depth-wise separable convolution head [45] with the encoder frozen (Light, less than 5% parameters). The optimal results were achieved across all three tiers, underscoring the significance of selecting an appropriate tuning method based on specific requirements.

**Pre-train**    We pre-trained Orochi-B (3D version) with the configuration listed in Table 6. The 2D version has a similar configuration, with slight differences on some setups (e.g. batch size). We have 2 two sets of pre-training devices. The A800 80Gx8 device is used for local pre-train and the H100 40Gx8 device is for streaming pre-train.

**Fine-tuning**    We followed the same setups as our code base for each task (see Section **??**), including the tuning resolution, epoch number, optimizer configurations and loss designs. The device we use for fine-tuning is NVIDIA 4090 24Gx4

Table 6: **Pretraining Configuration Parameters for Orochi-B**

| Category | Parameter | Value / Range and Description |
|---|---|---|
| Model / Encoder | `img_size` | (32, 224, 224) |
| | `patch_size` | 4 |
| | `pat_merg_rf` | 2 |
| | `in_chans` | 2 |
| | `embed_dim` | 128 |
| | `depths` | (4, 4, 4, 4) |
| | `drop_path_rate` | 0.2 |
| | `if_convskip` | True |
| | `out_indices` | (0, 1, 2, 3) |
| Mamba | `ssm_cfg` | None |
| | `norm_epsilon` | $1 \times 10^{-5}$ |
| | `initializer_cfg` | None |
| | `fused_add_norm` | True |
| | `rms_norm` | True |
| | `residual_in_fp32` | True |
| | `patch_norm` | True |
| | `use_checkpoint` | False |
| Decoder | `decoder_bn` | False |
| | `decoder_depthseparable` | False |
| | `decoder_head_chan` | 64 |
| Training | `batch_size` | 12 |
| | `lr` | 0.0005 |
| | `weight_decay` | 0.01 |
| | `warmup_ratio` | 0.1 |
| | `warmup_start_factor` | 0.01 |
| | `max_epoch` | 50 |
| | **Optimizer / Scheduler** | AdamW; WarmupCosine with cycles=0.5 |
| Deformation & Augmentation | Registration Flow Scaling | $\tanh(\cdot) \times 0.6$ (applied to the deformation field) |
| | Registration Gaussian Sigma Range | [1.5, 3.5] |
| | Perlin Noise Octaves | 4 |
| | Perlin Noise Persistence | 0.5 |
| | Mask Ratio | 0.5 |
| | Downsampling Scale Factor Range | [0.25, 0.75] |
| | Downsampling Noise Level Range | [0.01, 0.1] |
| | Downsampling Gaussian Sigma Range | [0.25, 1.0] |
| | Gaussian Noise Level Range | [0.075, 0.15] |
| | Salt vs. Pepper Ratio | 0.5 |
| | Salt & Pepper Noise Amount Range | [0.01, 0.05] |
| | Grid Image Parameters | Grid spacing = 4, Line width = 1 |

# C Extra Results

## C.1 Zero-shot Processing on Biomedical Images

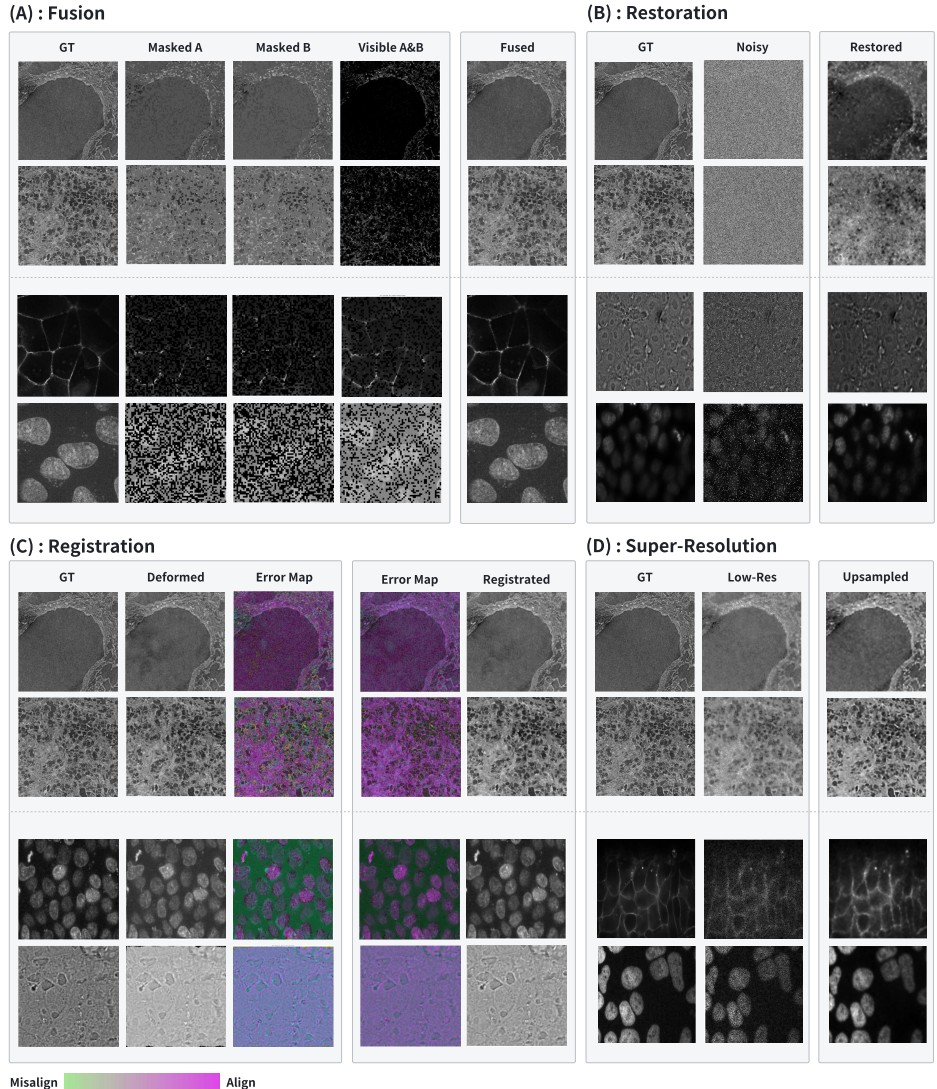

Figure 8: **Visualization of Orochi Zero-shot Processing to Different Degradation**. (A)-(D) panel shows the ability on four tasks respectively, with the dotted line separating medical images (top) and microscopy images (bottom).

In Figure 8, we present additional results demonstrating Orochi's zero-shot performance on both microscopy and medical images. Notably, Orochi yields satisfactory outcomes even when faced with extremely severe degradation, as exemplified in panel (B), row 2, and panel (C), rows 2 and 3.

## C.2 Registration & Fuison

**Fusion Model on Registration Task**   In Figure 9, we demonstrate that, despite the recent trend of pre-registration before fusion, these methods remain predominantly fusion-oriented and are not well-suited for addressing real-world registration tasks within the medical image registration community. However, Orochi represents a significant advancement in versatility, as it is designed not only for this specific scenario but also to achieve superior performance across all registration tasks.

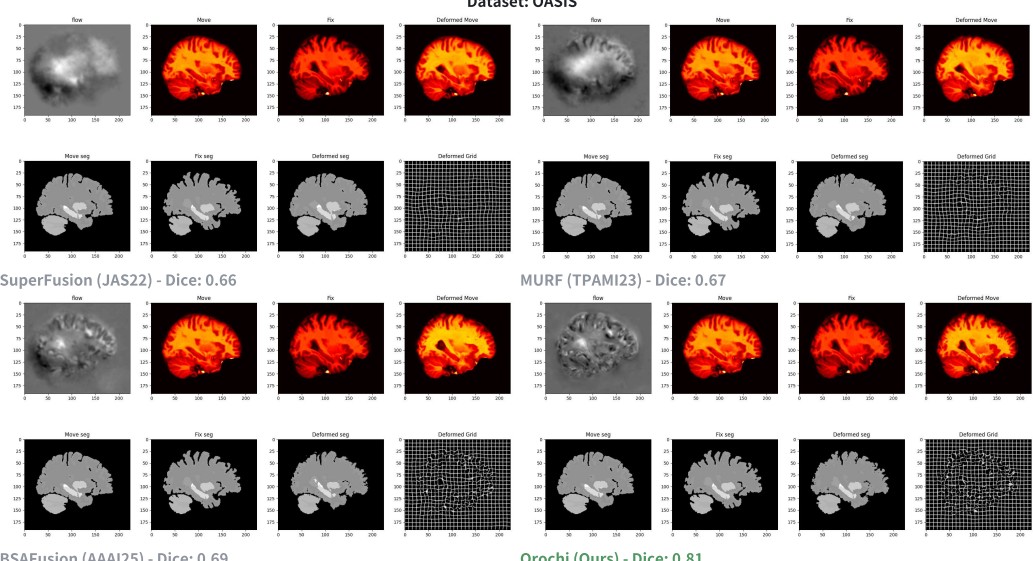

Figure 9: **Performance of Fusion-oriented methods on specialized registration benchmark.** We provide the inputs (Move, Fix), output (flow), and evaluation data (Move/Fix seg and Deformation Grid) for each visualization.

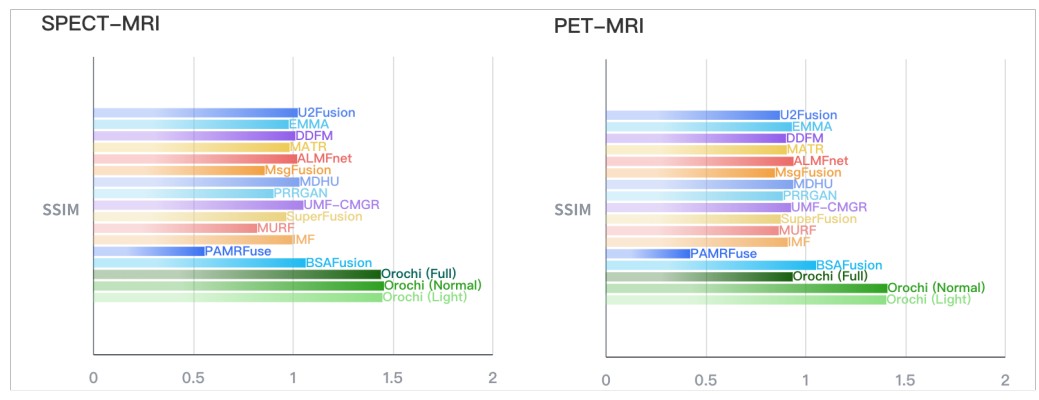

Figure 10: **Fusion Comparison with Brain SPECT&MRI / PET&MRI Data.**

**Patient to Atlas Brain Image Registration**    The regional deformation is learned unsupervised in Table 7. Only the raw image of the atlas and the patient's brain would be used for loss calculation while training. Then we evaluate the dice score between the segmentation maps of these two brains. Since Orochi is pre-trained in this unsupervised fashion, it shows excellent adaptation to this task, similar to the case with supervision.

**SPECT-MRI & PET-MRI Image Fusion**    In Figure 10, we performed comparative evaluations using state-of-the-art fusion techniques on two additional Harvard Whole Brain datasets obtained from https://www.med.harvard.edu/aanlib/. These datasets specifically focus on the fusion of SPECT and PET imaging with MRI. The results demonstrate that Orochi outperforms recent advancements such as BSAFusion and maintains superior efficiency.

Table 7: **Patient to Atlas Brain Registration Task.** During the training phase, the model aims to input paired MRI data from both the standard brain atlas and patient scans, to output a predicted registration flow. This flow is subsequently applied to the atlas data to compute the similarity between the registered atlas and the patient's scan. During the testing phase, the predicted flow is applied to the atlas brain segmentation map, and the Dice coefficient is evaluated against the patient's brain segmentation map.

| Method | Dice ↑ | % of $|J_\Phi| \leq 0$ |
|---|---|---|
| **Dataset: *IXI* [32]** | | |
| Affine | 0.386 ± 0.195 | — |
| SyN [61] | 0.645 ± 0.152 | ≤ 0.0001 |
| NiftyReg [62] | 0.645 ± 0.167 | 0.020 ± 0.046 |
| LDDMM [63] | 0.680 ± 0.135 | ≤ 0.0001 |
| deedsBCV [64] | 0.733 ± 0.126 | 0.147 ± 0.050 |
| VoxelMorph-1 [54] | 0.729 ± 0.129 | 1.590 ± 0.339 |
| VoxelMorph-2 [54] | 0.732 ± 0.123 | 1.522 ± 0.336 |
| VoxelMorph-diff [54] | 0.580 ± 0.165 | ≤ 0.0001 |
| CycleMorph [65] | 0.737 ± 0.123 | 1.719 ± 0.382 |
| MIDIR [66] | 0.742 ± 0.128 | ≤ 0.0001 |
| ViT-V-Net [67] | 0.734 ± 0.124 | 1.609 ± 0.319 |
| PVT [68] | 0.727 ± 0.128 | 1.858 ± 0.314 |
| CoTr [69] | 0.735 ± 0.135 | 1.292 ± 0.342 |
| nnFormer [70] | 0.747 ± 0.135 | 1.595 ± 0.358 |
| TransMorph-Bayes [32] | 0.753 ± 0.123 | 1.560 ± 0.333 |
| TransMorph-diff [32] | 0.594 ± 0.163 | ≤ 0.0001 |
| TransMorph-bspl [32] | 0.761 ± 0.122 | ≤ 0.0001 |
| TransMorph [32] | 0.754 ± 0.124 | 1.579 ± 0.328 |
| **Orochi** (Full) | **0.770 ± 0.120** | 1.592 ± 0.334 |
| **Orochi** (Normal) | 0.765 ± 0.121 | 1.571 ± 0.323 |
| **Orochi** (Light) | 0.752 ± 0.126 | 1.499 ± 0.301 |

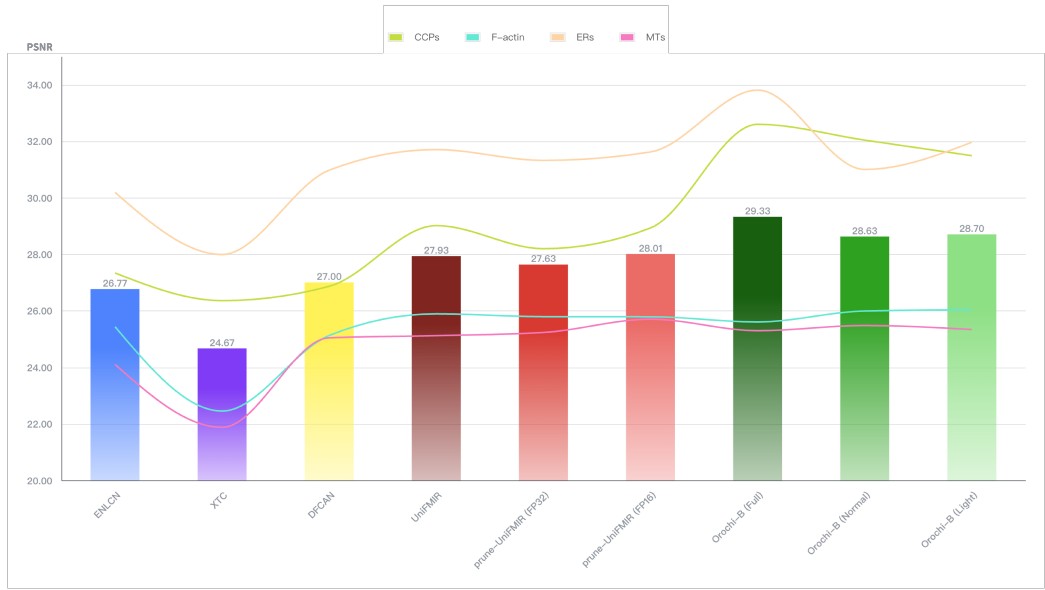

Figure 11: **Stress Test on BioSR Benchmark.** We trained Orochi on four subsets concurrently, thereby reducing the cost associated with hyperparameter searching. The results were compared against baselines that involved separate hyperparameter searches for each subset. The bar chart illustrates the average PSNR values, while the lines, colour-coded according to the legends, indicate the performance metrics for each respective subset.

## C.3 Super-Resolution & Restoration

**Stress test on joint multi-modal data image repairing**  In this additional validation, we aim to evaluate Orochi's performance under stress using an extended benchmark. The BioSR [71] benchmark comprises four distinct categories of microscopy image pairs (x2 low/high imaging quality), captured by a multimodal structured illumination microscopy (SIM) system, encompassing Clathrin-Coated Pits (CCPs), Endoplasmic Reticula (ERs), Microtubules (MTs), and F-actin Filaments. Specifically, Orochi was trained on all four datasets concurrently, whereas the baseline [72, 73, 74, 6] models were trained separately on each dataset. This deliberate approach highlights Orochi's capability in resource-constrained environments, where conducting hyperparameter searches for each subset is not feasible. As illustrated in Figure 11, despite the training constraints imposed on Orochi, an absolute improvement is still observed, further demonstrating its capability and efficiency.

## D  Limitations

Two limitations in our paper remain unaddressed at present. First, due to constraints on computational resources and group size, we were unable to further investigate the scaling law of our method during pre-training. This limitation also indicates that our focus was restricted to low-level tasks as presented in the paper. However, we firmly believe that a unified model for life sciences, capable of excelling in both high-level understanding tasks and low-level generation tasks, will emerge in the future. This is also an emerging trend that has already demonstrated progress in general applications.

