# OpenReview forum: "Orochi: Versatile Biomedical Image Processor"
_NeurIPS.cc/2025/Conference — NeurIPS 2025 spotlight_

### Official Review · Reviewer_Z17s · 2025-06-25

**Clarity:** 3
**Significance:** 2
**Originality:** 2
**Rating:** 3
**Confidence:** 4

**Summary:**

A foundation model on biomedical image proposing is introduced in this draft. Multiple pre-training techniques are assembled to learn from 100+ datasets. Fine tuning is applied to the foundation model (Mamba + Swin-Transformer) for various downstream tasks such as restoration, super-resolution, fusion, and registration. The proposed model achieves comparable or superior performance compared to other task-specific models.

**Questions:**

A long list of pre-training techniques are available in the community. Four pre-training techniques are assembled in this draft. The underlying motivations to choose these four can be discussed with more details and analyzed with more experiments.

The backbone model uses Mamba and Swin-Transformer. Though they are not the new contributions in the draft, the backbone and fine-tuning can be briefly described in the main method section, rather than putting them entirely in the supplemental materials. A lot of engineering details in the current method section can be moved to supplement.

There are 100+ datasets with various tasks. When training the model, are all of them used at once for multi-task learning? Or, are a subset of them randomly sampled in each epoch? Or, is each of them iterated during the training? Some details on the training strategies can be added.

**Ethical Concerns:**

["NO or VERY MINOR ethics concerns only"]

**Final Justification:**

The reviewer has two major doubts on the paper:
1. Biomedical images are very domain-specific. It's hard for the reviewer to think why the information learned from one domain (e.g., a microscopy image) could be useful for the other domain with big difference (e.g., MRI brain images).
2. Learning from unrelated images could be another issue. If we can learn from multi modal data from the same patient regarding a certain type of disease (e.g., CT, lab test, other electronical health records, etc), it could be useful to the diagnosis, like what MedGemma did.
Due to the concerns, I remain with the reject decision.

**Limitations:**

No discussion in the paper.

**Paper Formatting Concerns:**

Some major method descriptions are left in the supplemental materials.

**Quality:**

2

**Strengths And Weaknesses:**

Strength:

+ This paper is a demonstration of the scaling law. Using a lot of data for the pre-training and then fine-tune it for various tasks;

+ Collecting 100+ datasets to build a model involves great effort;

Weakness:

- Biomedical image modalities can be quite different. The reader is not clear why features learned on one modality (e.g., microscopy cell images) can be used for the other modality with a large domain difference (e.g., MRI on brain images);

- Some biomedical image tasks requires paired images of different modalities on the same object. As the collected 100+ datasets are quite heterogeneous, how can this problem be solved?

- The method section is focused on the engineering implementation details. There is no new technique introduced.

---

> ### Author Rebuttal · Authors · 2025-07-28
>
> First and foremost, we would like to thank the reviewer for recognizing the **substantial effort** invested in our work and for the careful and diligent review. We **do include a Limitations section** in our manuscript, and here we will provide point-by-point responses to **all comments**. We aim to elaborate on aspects that were condensed due to page limitations and hope to earn your **further approval** by addressing your questions. Thank you!
>
> ***
>
> **Q1:**
> > Biomedical image modalities can be quite different. The reader is not clear why features learned on one modality (e.g., microscopy cell images) can be used for the other modality with a large domain difference (e.g., MRI on brain images).
>
> **A1:**
> As we **articulated in the first paragraph** of our introduction, the primary processing challenges for biomedical images, compared to natural images, stem from two main aspects. First is the issue of imaging device operational trade-offs: long imaging times can damage the subject, while shorter times may compromise image quality. This trade-off is a universal challenge present in **both microscopy and clinical imaging**. The second is the shortcomings of imaging modalities. For instance, MRI is superior for soft tissue imaging, while CT excels at imaging hard structures; many such **complementary modalities exist.**
>
> Therefore, **different imaging modalities are not entirely disconnected**. They share a degree of commonality and complementarity, from their underlying imaging principles to their imaging content. This understanding motivated our decision to broadly increase the diversity of our training data.
>
> Furthermore, our experiments demonstrate that a model pre-trained in this manner exhibits superior efficiency and performance when fine-tuned on various downstream tasks. This not only validates our hypothesis but also aligns with the broad consensus in the general vision domain that pre-training on diverse datasets like ImageNet leads to better performance for foundation models.
>
> ***
>
> **Q2:**
> > Some biomedical image tasks require paired images of different modalities on the same object. As the collected 100+ datasets are quite heterogeneous, how can this problem be solved?
>
> **A2:**
> We understand your concern. While downstream tasks like registration and fusion often require paired images, collecting such data at a large scale for pre-training is challenging. To address this, we developed **targeted self-supervised strategies**:
> * **For Registration:** We create a proxy task by applying multi-scale non-rigid deformations to an original image. The model's pre-training objective is to generate the registration grid that reverses this deformation. When fine-tuning on a downstream task, which may have only a small number of paired samples, our model can leverage this pre-learned ability to generate registration grids. This helps it adapt faster and more effectively, mitigating the risk of overfitting that models without such pre-training would face.
> * **For Fusion:** We build upon existing work that uses Masked Image Modeling (MIM) for fusion pre-training **[1]**. We advance this by employing a **dual-masking** strategy. This forces the model to first learn to fuse two non-overlapping masked regions of an image before predicting the content of the remaining masked area. Our ablation studies confirm that this method yields superior results compared to a standard MIM pre-training approach.
>
> In summary, our pre-training phase does not require strictly paired data; instead, it learns through purely self-supervised methods. This pre-training paradigm allows us to scale up our dataset size and simplifies the process of incorporating new, unpaired data in the future.
>
> ***
>
> **Q3:**
> > The method section is focused on the engineering implementation details. There is no new technique introduced. (Weakness # 3)
>
> > A lot of engineering details in the current method section can be moved to the supplement. (Question # 2)
>
> **A3:**
> While we did describe our self-supervised strategies in detail in the Method Section, **we respectfully disagree with the assessment that "There is no new technique introduced."**
>
> Our writing style was intentionally modeled after widely recognized papers that introduced influential self-supervised strategies, such as MAE **[2]** (whose Method Section details its mask ratio, mask token) and I-JEPA **[3]** (which follows a similar pattern). Our paper, in fact, features a more complex design of self-supervised strategies, and detailed descriptions are unavoidable.
>
> Specifically, our novelty lies in several areas:
> * Our Task-related Joint-embedding Pre-training (TJP) is a novel approach that simultaneously utilizes specific degradation types across multiple biomedical, domain-specific tasks as self-supervised learning strategies. To the best of our knowledge, this concept has not been previously proposed in existing literature.
> * Within each self-supervised strategy, we introduced new considerations. For example, we extended single-mask image modeling **[1]** to a dual-masking approach to better suit fusion tasks. We also improved upon the single-scale image degradation scheme in DeepLP **[4]** by implementing a multi-scale version.
> * Our Methods section also covers the key concept of our designs for data acquisition, model architecture, and post-training strategies. While we are not the first to use a Swin-Mamba framework, we are the first to apply it to this breadth of multi-task pre-training. Furthermore, our three-tiered fine-tuning strategy and its validation for different scenarios are also contributions. We will revise the manuscript to highlight these innovations more prominently in the main text in a future version with a more generous page limit.
>
> While these designs certainly build upon the work of others, we believe the claim of "no new technique" is not a fair characterization, especially when compared to other papers that focus on advancements in self-supervised strategies.
> ***
> **Q4:**
> > A long list of pre-training techniques are available in the community. Four pre-training techniques are assembled in this draft. The underlying motivations to choose these four can be discussed with more details and analyzed with more experiments.
>
> **A4:**
> We chose these four pre-training techniques (for restoration, super-resolution, registration, and fusion) not only because they represent the four primary low-level processing tasks in biomedical imaging, but more importantly, because of their deep **underlying relationships**, which we took considerable space to explain.
>
> The need to preprocess biomedical images stems from two fundamental inadequacies of current imaging methods:
> 1.  **Shortcomings of imaging modalities:** A single modality often has limitations. Therefore, a model should learn to **fuse** information from different modalities, even if they are not perfectly aligned (which requires **registration**).
> 2.  **Imaging device operational trade-offs:** Even with an adequate modality, we cannot guarantee high image quality. Thus, we want the model to extract useful information even from low signal-to-noise ratio images and perform image reconstruction (**restoration** and **super-resolution**).
>
> In our ablation studies, we demonstrated that if our TJP is removed and replaced with standard MIM methods, the performance suffers, especially on spatially-aware tasks like registration. We have additionally performed new ablations comparing TJP to DINOv2 **[5]** pre-training. As shown in the results below, the TJP approach remains more robust.
>
> |  | OASIS (Registration) | VIFB (Fusion) | CARE (Restoration) | HBA (Super-Resolution) |
> | :--- | :--- | :--- | :--- | :--- |
> | Strategy | Dice | Qabf | PSNR | PSNR |
> | MAE (single mask) | 71.22 | 0.36 | 26.67 | 29.17 |
> | I-JEPA (dual mask) | 69.97 | 0.39 | 25.02 | 28.81 |
> | DINOv2 (multi-augment) | 81.12 | 0.40 | 28.17 | 31.01 |
> | Orochi (task multi-augment) | **83.62** | **0.41** | **29.88** | **33.63** |
> ***
> **Q5:**
> > There are 100+ datasets with various tasks. When training the model, are all of them used at once for multi-task learning? Or, are a subset of them randomly sampled in each epoch? Or, is each of them iterated during the training? Some details on the training strategies can be added.
>
> **A5:**
> Thank you for the suggestion. Yes, **all of the studies were included** for multi-task learning, but not all the raw data was used. It is important to clarify that the original studies associated with those data did not necessarily involve deep learning, meaning properties like data dimensionality and resolution may not be directly suitable for training a deep learning model.
>
> To leverage this diverse data, our solution involved several steps. After an initial filtering phase (to remove non-imaging data like tables and graphs), we performed multi-scale random sampling on the source data (as detailed in Section 3.2 of our paper). We then employed a hybrid training strategy using both locally stored (fixed patch/volume) and streamed data (random patch/volume).
> ***
> **Reference**
>
> **[1]**  Li, Jiayang, et al. "MaeFuse: Transferring omni features with pretrained masked autoencoders for infrared and visible image fusion via guided training." IEEE Transactions on Image Processing (2025).
>
> **[2]** He, Kaiming, et al. "Masked autoencoders are scalable vision learners." Proceedings of the IEEE/CVF conference on computer vision and pattern recognition. 2022.
>
> **[3]** Assran, Mahmoud, et al. "Self-supervised learning from images with a joint-embedding predictive architecture." Proceedings of the IEEE/CVF Conference on Computer Vision and Pattern Recognition. 2023.
>
> **[4]** Fang, Linjing, et al. "Deep learning-based point-scanning super-resolution imaging." Nature methods 18.4 (2021): 406-416.
>
> **[5]** Oquab, Maxime, et al. "Dinov2: Learning robust visual features without supervision." arXiv preprint arXiv:2304.07193 (2023).

---

> > ### Author Response · Authors · 2025-08-03
> >
> > ### Dear Reviewer,
> >
> > We are writing to follow up on our previous rebuttal.
> >
> > We would like to kindly mention that we have had a successful and thorough discussion with **Reviewer eeYn**, and we were able to resolve all of their concerns.
> >
> > Our rebuttal, which we prepared with considerable effort to address your insightful questions, is intended to facilitate a positive and comprehensive discussion with you. We believe the materials and new experiments directly address the points you raised.
> >
> > We look forward to your feedback and greatly appreciate the time you have dedicated to reviewing our work.
> >
> > Best regards,
> >
> > The Authors

---

> > ### Comment · Reviewer_Z17s · 2025-08-06
> > **The reviewer appreciated the response from the authors.**
> >
> > Biomedical images are very different from natural images, as the image formation processes of different modalities are so different. The reviewer is not convinced about the motivation of learning from a model (e.g., pretrained on a pathological image dataset) and fine-tuning it on MRI brain images. There are so many domain-specific biomedical images and tasks, and the problem formulation is doubtful to the reviewer.

---

> > > ### Author Response · Authors · 2025-08-06
> > > **Further Discussion on the motivation**
> > >
> > > ### Dear Reviewer,
> > >
> > > We agree that biomedical images differ from natural images. This is precisely why we did not simply adopt models pre-trained on natural images (e.g., MAE, I-JEPA, DINOv2). Instead, we invested significant effort to pre-train a new model, Orochi, from the ground up. Our dataset encompasses image data from **hundreds of biomedical studies** (fluorescence microscopy, Digital pathology imaging, Yeast studies, High-content screening, etc.) across Image Data Resources (IDR), Human Induced Pluripotent Stem Cell Atlas (HIPSC), and Human Organ Atlas (HIPCT), covering a wide array of common imaging modalities (Light Sheet Microscopy, Multimodal Structured Illumination Microscopy, CT, SPECT, PET, MRI, etc.), which also includes the MRI and pathological images you mentioned. (see **Data_Sources.xlsx** file located in our Supplementary Materials, within the "dataset" folder, for more information.)
> > >
> > > Therefore, we wish to clarify that our focus is on the **"foundation model"** approach: ***"enabling the model to learn the underlying relationships and process a vast spectrum of modalities through self-supervised pretraining."***  This is distinct from a "continual learning" approach, which would focus more on "first train on a few modalities, then transfer to new ones."
> > >
> > > Furthermore:
> > >
> > > * In **A1**, we elaborated on the rationale behind the assertion that "different imaging modalities are not entirely disconnected." To substantiate this claim, we provided examples, such as Lightsheet Microscopy and CT, which encounter similar operational trade-offs for imaging time vs. quality. Furthermore, CT and MRI are employed in a complementary manner to address their individual limitations in visualizing both soft tissues and hard anatomical structures.
> > >
> > > * In **A4**, we elaborated on how these underlying inter-modal relationships motivated our Task-related Joint-embedding Pre-training (TJP) design, and we provided a comparison with other pre-training methods.
> > >
> > > To conclude, your observation that “There are so many domain-specific biomedical images and tasks” perfectly captures the problem we aim to solve. The current fragmented landscape makes it difficult for biologists to choose from a **"toolbox" of specialized models**. Orochi is proposed as a **"Swiss Army knife"** to lower this barrier, illustrating the promise of a versatile, multitask foundation model for the community. This core motivation was **recognized by all the other reviewers**, a consensus we hope you might also consider.
> > >
> > > Should any of your concerns about our rebuttal remain, we would be grateful if you could articulate them specifically. We stand ready to provide a prompt and detailed response.
> > >
> > > Thank you for your time and consideration.
> > >
> > > Sincerely,
> > >
> > > The Authors

---

> > > > ### Author Response · Authors · 2025-08-07
> > > > **Follow-up on our previous reply**
> > > >
> > > > ### Dear Reviewer,
> > > >
> > > > Since all the other reviewers have conducted their final score, as a follow-up on our previous reply, we hope we have clarified that the essence of our work is a foundation model designed to encounter and process a wide variety of modalities through self-supervised learning during the pre-training stage, rather than a project focused on continual learning.
> > > >
> > > > To further demonstrate the utility of our pre-training approach, we have conducted an additional, urgent experiment. The task is MRI Axial Super-resolution on the HBA dataset, as referenced in Table 2 of our manuscript. Using the same model architecture (MambaUNet), we compared the following four setups:
> > > > 1.  **Training from scratch** using only MRI data.
> > > > 2.  **BioSR (Microscopy data) pre-training**, followed by fine-tuning on MRI data.
> > > > 3.  **BioSR + MRI pre-training**, followed by fine-tuning on MRI data.
> > > > 4.  **Loading the Orochi checkpoint** (pre-trained on a vast spectrum of modalities) and fine-tuning on MRI data.
> > > >
> > > > The results are as follows:
> > > >
> > > > **MambaUNet Architecture**
> > > > | Method | PSNR (4mm) | SSIM (4mm) | PSNR (8mm) | SSIM (8mm) |
> > > > | :--- | :--- | :--- | :--- | :--- |
> > > > | MRI from scratch | 28.27 | 0.90 | 27.98 | 0.84 |
> > > > | BioSR pre-train & MRI Finetune | 29.01 | 0.92 | 28.16 | 0.82 |
> > > > | BioSR+MRI pre-train & MRI Finetune | 29.74 | 0.93 | 28.88 | 0.84 |
> > > > | **Orochi pre-train & MRI Finetune** | **35.33** | **0.95** | **31.93** | **0.89** |
> > > >
> > > > As the results clearly show, Orochi's large-scale pre-training achieves a substantial performance lead. It is also noteworthy that even without employing any continual learning techniques, all pre-training methods outperform training solely from scratch on MRI data. This supports our hypothesis that even across different imaging modalities, there are similarities in low-level degradation patterns. In fact, we observed this phenomenon at the very beginning of our project, which is what motivated us to scale up the pre-training to explore the possibility of a unified biomedical image processor for low-level tasks.
> > > >
> > > > We hope that this new experiment, in conjunction with our previous rebuttal, provides a satisfactory answer to your concerns.
> > > >
> > > > Sincerely,
> > > >
> > > > The Authors

---

> > > ### Author Response · Authors · 2025-08-08
> > > **Kindly request for feedbacks**
> > >
> > > ### Dear Reviewer,
> > >
> > > As the discussion period will close in **less than 1 day**, we wanted to kindly follow up and ask if you would have a moment to provide brief feedback on our rebuttal.
> > >
> > > We dedicated a great deal of effort to addressing your concerns, driven by our deep respect for your feedback and suggestions. We believe that responsive dialogue is crucial for a fair and effective review process, a principle we know is central to the NeurIPS community. We would be very grateful for the opportunity to ensure all your points have been resolved.
> > >
> > > Thank you for your understanding and time.
> > >
> > > Best regards,
> > >
> > > The Authors

---

### Official Review · Reviewer_eeYn · 2025-06-28

**Clarity:** 3
**Significance:** 3
**Originality:** 3
**Rating:** 4
**Confidence:** 3

**Summary:**

Orochi is introduced as the first versatile biomedical image processor designed for diverse low-level tasks like restoration, super-resolution, registration, and fusion, overcoming limitations of task-specific specialist models. It employs a novel Task-related Joint-embedding Pre-Training (TJP) on a large, multi-scale dataset and utilizes an efficient Multi-head Hierarchy Mamba architecture. Experiments demonstrate that Orochi achieves state-of-the-art or competitive performance across tasks, even with parameter-efficient fine-tuning, highlighting the effectiveness of its data and pre-training strategies.

**Questions:**

Please refer to the weakness.

**Ethical Concerns:**

["NO or VERY MINOR ethics concerns only"]

**Final Justification:**

The authors have adequately addressed all of my questions and concerns. Therefore, I will maintain my current score.

**Limitations:**

Yes.

**Paper Formatting Concerns:**

None.

**Quality:**

3

**Strengths And Weaknesses:**

Strengths

1. Orochi, a biomedical image processing model capable of handling various bio-imaging modalities and low-level tasks, addresses practical challenges faced by medical researchers and proposes a solution that can significantly reduce their computational and operational costs.

2. The model is trained on a large-scale dataset (over 100 terabytes), positioning it as a promising foundation model in the biomedical domain.

3. The effectiveness of the proposed method is validated through comprehensive performance comparisons with a variety of existing models.

Weakness

1. Limited detailed information on the diversity and balance of the pre-training dataset: While the paper highlights the large scale (100+ public studies, 100TB+ of data), it lacks clarity on the dataset’s composition—such as imaging modalities, species, pathological vs. normal cases, resolution range, and artifact types. This makes it difficult to assess how broadly applicable the claimed versatility of Orochi truly is.

2. Failure cases are not discussed, making it difficult to understand the model’s limitations in real-world scenarios.

---

> ### Author Rebuttal · Authors · 2025-07-27
>
> We sincerely thank the reviewer for the **recognition of the motivation, data selection, model training, and experimental presentation** of our work. In the following, we provide point-by-point responses to **all comments**, elaborating on areas that we were unable to detail due to space constraints. We hope this addresses your questions and earns your **further approval**. Thank you again!
>
> ***
>
> **Q1:**
> > Limited detailed information on the diversity and balance of the pre-training dataset: While the paper highlights the large scale (100+ public studies, 100TB+ of data), it lacks clarity on the dataset’s composition—such as imaging modalities, species, pathological vs. normal cases, resolution range, and artifact types. This makes it difficult to assess how broadly applicable the claimed versatility of Orochi truly is.
>
> **A1:**
> We invite you to review the **`Data_Sources.xlsx`** file located in our **Supplementary Materials**, within the **"dataset"** folder. This file contains the **organized metadata** that covers most of the information you mentioned (modality, species, dimension, resolution, directory...). Our **data diversity** is primarily derived from the modality diversity of the Image Data Resource (IDR) **[1]** website (which includes Light-sheet fluorescence microscopy, Digital pathology imaging, Yeast studies, High-content screening, etc.) and is supplemented by Atlas datasets (such as the Human Organ Atlas **[2]** and the Human Induced Pluripotent Stem Cell Atlas **[3]**). Due to page limitations, the main body of our paper focused on **task versatility**, with many details on data versatility expanded upon in the appendix. We hope this clarification resolves your concerns.
>
> ***
>
> **Q2:**
> > Failure cases are not discussed, making it difficult to understand the model’s limitations in real-world scenarios.
>
> **A2:**
> Like all scientific papers, our work has its limitations, and we have **already included a "Limitations" section** in the manuscript. The nature of our training methodology and data types means that our model is **primarily focused on low-level tasks**. We have since conducted additional experiments by fine-tuning the model on two high-level tasks, namely segmentation and classification.
>
> First, we conducted new tests on the model's performance on a subset of BTCV **[4]**, REFUGE **[5]**, and CadVidSet **[6]** segmentation datasets. We strictly adhered to the training hyperparameter settings of MedSAM2 **[7]** and carried out a complete fine-tuning following its GitHub repository. Baseline results are from the original paper.
>
> | Model | REFUGE-2D | CadVidSet-2D | BTCV-Aorta-3D | BTCV-Liver-3D |
> | :--- | :--- | :--- | :--- | :--- |
> | TransUNet | 0.363 | 0.316 | **0.920** | 0.969 |
> | Swin-UNet | 0.289 | 0.226 | 0.892 | **0.975** |
> | nnUNet | 0.349 | 0.418 | 0.877 | 0.948 |
> | SAM2 | 0.558 | 0.539 | 0.835 | 0.861 |
> | MedSAM2 | 0.799 | **0.861** | 0.896 | 0.865 |
> | Orochi | **0.811** | 0.857 | 0.911 | 0.950 |
>
> Also, we conducted classification tasks on TissueMNIST (2D) and OrganMNIST (3D) datasets. All experimental setups are from the MedMNIST series papers **[8] [9] [10]**.
> ### Dice Score
> | Methods (TissueMNIST-2D-224^2) | AUC | ACC | Methods (OrganMNIST-3D-28^3) | AUC | ACC |
> | :--- | :--- | :--- | :--- | :--- | :--- |
> | ViT-Base | 0.948 | 0.729 | ResNet-18+3D | **0.996** | **0.907** |
> | MAE-Base | 0.945 | 0.719 | ResNet-50+3D | 0.994 | 0.883 |
> | CLIP-Base | 0.922 | 0.662 | ResNet-50+2.5D | 0.974 | 0.769 |
> | EVAv2-Base | 0.940 | 0.703 | Auto-sklearn | 0.977 | 0.814 |
> | DINOv2-Base | 0.944 | 0.715 | AutoKeras | 0.979 | 0.804 |
> | Orochi | **0.950** | **0.733** | Orochi | 0.995 | 0.891 |
>
> We have demonstrated Orochi's capability to adapt to high-level tasks; however, we intend to avoid exaggerating its scope of application. We expect that its performance may decline to average or insufficient levels when applied to a broader range of high-level tasks, as our primary objective remains centred on the development of a unified low-level processor, as originally stated since the abstract of this paper.
> ***
> **Reference**
>
> **[1]**  Williams, Eleanor, et al. "Image Data Resource: a bioimage data integration and publication platform." Nature methods 14.8 (2017): 775-781.
>
> **[2]** Walsh, Claire L., et al. "Imaging intact human organs with local resolution of cellular structures using hierarchical phase-contrast tomography." Nature methods 18.12 (2021): 1532-1541.
>
> **[3]** Viana, Matheus P., et al. "Integrated intracellular organization and its variations in human iPS cells." Nature 613.7943 (2023): 345-354.
>
> **[4]** Landman BA, Xu Z, Igelsias JE, Styner M, Langerak TR, and Klein A, "MICCAI multi-atlas labeling beyond the cranial vault workshop and challenge," (2015)
>
> **[5]** Orlando, José Ignacio, et al. "Refuge challenge: A unified framework for evaluating automated methods for glaucoma assessment from fundus photographs." Medical image analysis 59 (2020): 101570.
>
> **[6]** Wang, Lu, et al. "Coronary artery segmentation in angiographic videos utilizing spatial-temporal information." BMC medical imaging 20.1 (2020): 110.
>
> **[7]** Zhu, Jiayuan, et al. "Medical sam 2: Segment medical images as video via segment anything model 2." arXiv preprint arXiv:2408.00874 (2024).
>
> **[8]** Yang, Jiancheng, et al. "Medmnist v2-a large-scale lightweight benchmark for 2d and 3d biomedical image classification." Scientific Data 10.1 (2023): 41.
>
> **[9]** Yang, Jiancheng, Rui Shi, and Bingbing Ni. "Medmnist classification decathlon: A lightweight automl benchmark for medical image analysis." 2021 IEEE 18th International Symposium on Biomedical Imaging (ISBI). IEEE, 2021.
>
> **[10]** Doerrich, Sebastian, et al. "Rethinking model prototyping through the MedMNIST+ dataset collection." Scientific reports 15.1 (2025): 7669.

---

> ### Comment · Reviewer_eeYn · 2025-08-02
>
> Thank you for your detailed response. Most of my concerns have been resolved. However, I still have one remaining question. Could you provide more details on the failure cases in low-level tasks beyond what is currently discussed in the Limitations section? The section as it stands does not appear to cover specific failure examples. Since it may be difficult to present them qualitatively, describing representative situations or providing quantitative evidence would also be helpful.

---

> > ### Author Response · Authors · 2025-08-02
> > **Further Discussion on Limitations**
> >
> > Thank you for acknowledging that most of your concerns have been resolved. We are very happy to continue the discussion on the final point regarding "details on the failure cases in low-level tasks."
> >
> > First, similar to the failure cases we observed in high-level task adaptation with the OrganMNIST dataset (as mentioned in our main rebuttal), we have found that in low-level tasks, the benefits of our pre-training could diminished if the processing resolution deviates significantly from our pre-training resolution of `224x224` (for 2D) or `224x224x32` (for 3D).
> >
> > For instance, when performing restoration on the CARE dataset, which consists of patches from a single large microscopy image, we attempted to increase the processing patch resolution to `896x896` or higher. The goal was to reduce patch boundary inconsistencies. However, we observed that this led to a decrease in the overall PSNR metric:
> >
> > | Method | PSNR (XY) | PSNR (XZ) |
> > | :--- | :--- | :--- |
> > | 224*224 | 28.31 | 28.52 |
> > | 896*896 | 26.79 | 27.11 |
> >
> > In practice, there are specialized methods to address such edge inconsistency issues in microscopy image restoration, with the simplest being the use of sliding, overlapping patches during the patchifying process **[1]**. However, we wish to extend this point to a broader observation: in various low-level biomedical image tasks, many similar, nuanced design considerations require attention (e.g., edge inconsistency in restoration, the pursuit of diffeomorphism in registration **[2]**). Our current work cannot comprehensively cover all these fine-grained details.
> >
> > In summary, Orochi represents our current attempt to demonstrate the feasibility of a versatile, multi-task model to the community. We will continue to explore frameworks in our future work that are both general-purpose and compatible with such detailed processing. We hope this response earns your approval. Thank you once again!
> >
> > **References**
> >
> > [1] Isensee, F., Jaeger, P.F., Kohl, S.A.A. et al. nnU-Net: a self-configuring method for deep learning-based biomedical image segmentation. *Nat Methods* 18, 203–211 (2021).
> >
> > [2] Li, L., Li, L., Zhang, Y. et al. Cyclic deformable medical image registration with prompt: deep fusion of diffeomorphic and transformer methods. *Appl Intell* 55, 296 (2025).

---

> ### Comment · Reviewer_eeYn · 2025-08-02
>
> Thank you for your response. I believe Orochi is a valuable contribution, especially in its focus on generality and multi-task capability. All of my concerns have been resolved, and I will maintain my current score.

---

### Official Review · Reviewer_Lxhu · 2025-07-02

**Clarity:** 3
**Significance:** 3
**Originality:** 3
**Rating:** 5
**Confidence:** 5

**Summary:**

This paper pretrains a foundation model for low-level biomedical image processing tasks, including restoration, super-resolution, registration and fusion. Because of the high heterogeneity of biomedical images (regarding image dimension, modality, imaging subjects and body parts, etc,), there are a large body of methods dealing with different tasks in different settings and it is usually challenging for researcher to find the proper tool for there specific need. Therefor, it is valuable to develop a unified model that can perform different tasks in different settings. This paper collect a large body of biomedical data from 100 studies, and develops a self-supervised pre-training strategy suitable for low-level image processing task. The pretrained model is then fine-tuned and compared to a set of specialist models on each task and achieves comparable or superior performance.

**Questions:**

please respond to the weakness points

**Ethical Concerns:**

["NO or VERY MINOR ethics concerns only"]

**Final Justification:**

My major concern about the insufficient validation has been clarified.

**Limitations:**

Yes

**Quality:**

3

**Strengths And Weaknesses:**

Strength
1.The motivation is reasonable and it is valuable to develop a unified model that can be applied to different biomedical image processing tasks.

2.The paper is well written and easily to follow.

3.This study collected and curated a relatively large body of biomedical data for pretraining.

4.The proposed pre-training strategy seems suitable for low-level processing tasks.


Weakness
1.Actually, the practical value of low-level tasks is relatively low, especially when comparing to high-level tasks, such as classification, segmentation, detection. Though there are commonly used metrics for evaluating low-level tasks, their practical meaning is more demonstrated in downstream high level tasks. For example, image fusion methods are usually evaluated in downstream detection/classification tasks.

2.For each of the four tasks, evaluation is only performed on one or very small number of datasets. We know that because of the high heterogeneity of biomedical images, the performance on one dataset can hardly generalize to all other datasets with slightly different settings. The evaluation on each task is far from being enough to show that the proposed method is generally applicable for the specific task type under different settings.

3.For the image fusion task, only three metrics are used to compare different methods, which is too small. Current image fusion studies typically utilize much more metrics.

4.The training strategies of the proposed methods and the baselines are different in many experiments, which may make the comparison unfair.

5.For the image registration task, there are a large set of different setting which bring different kinds of difficulty. Such as, registration between 2D images or 3D volumes, registration between different modality pairs or within the same modality, registration of different body parts with significant or not deformation, etc. Much more experiments are needed to conclude that a model can be generally applied to image registration task.

---

> ### Author Rebuttal · Authors · 2025-07-29
>
> We thank the reviewer for recognizing that our **motivation is reasonable**, the paper is **well-written and easy to follow**, and that our **training methodology, data usage, and workload are appropriate** for the topic. Despite the divergence in the final score, we would like to provide a detailed point-by-point response to address your concerns and hopefully resolve any outstanding questions.
> ***
> **Q1:**
> > Actually, the practical value of low-level tasks is relatively low, especially when comparing to high-level tasks. Though there are commonly used metrics for evaluating low-level tasks, their practical meaning is more demonstrated in downstream high level tasks.
>
> **A1:**
> We agree that high-level tasks are a vital part of AI for Life Science, but we **do not agree this supports the conclusion that low-level tasks are subordinate to high-level ones**. Purely low-level processing tasks command significant attention within the community. For eaxample, starting from early efforts like Content-aware image restoration (CARE) **[1]**, which focused on fluorescence microscopy restoration and was published in *Nature Methods* in 2018, subsequent works such as BioSR **[2]** (Dataset & Benchmark), DeepLP **[3]** (Self-Supervision Learning), and UniFMIR **[4]** (Foundation Model) have repeatedly appeared in top-tier journals, each with a distinct focus.
>
> From a presentation standpoint, a paper should have a clear focus. As we have emphasized since the title, our goal is to develop a versatile biomedical image **processor**, not an **analyzer**. Our work is benchmarked against low-level foundation models like UniFMIR in all aspects, including writing style, experimental setup, and validation methods. In these papers, using **indirect downstream tasks to validate performance is not "usual"** and can be **less intuitive**, as the choice of the downstream model can influence a fair comparison of the upstream processing effects. If you could specify which articles you would like us to compare against, we would be more than happy to make improvements.
>
> Furthermore, while works focused on high-level tasks might emphasize downstream performance (e.g., the recent Cellpose 3 **[5]**, which integrates a restoration module to aid segmentation), these are not the most suitable for a direct comparison with our model. Nevertheless, we performed an additional comparison using the same noisy dataset (68 images) from their work and fine-tuned our model on Cellpose3's training set. For further details on our model's performance on high-level tasks, we invite you to see our response to Reviewer S4FJ's Q7, which includes our fine-tuning results on segmentation and classification. We hope this statement clarifies the focus of our work. Thank you for your contribution!
> | Methods | AP@0.5 ↑ | AP@0.75 ↑ |
> | :--- | :--- | :--- |
> | Noisy Image | 0.41 | 0.23 |
> | Noise2void | 0.50 | 0.26 |
> | Noise2self | 0.51 | 0.28 |
> | Cellpose3 | 0.69 | 0.41 |
> | Orochi+Cellpose | **0.71** | **0.44** |
> ***
>
> **Q2:**
> > For each of the four tasks, evaluation is only performed on one or very small number of datasets. The evaluation on each task is far from being enough to show that the proposed method is generally applicable.
>
> **A2:**
> We have included **13 datasets**, covering over **30 baselines** for comparison. At least **4 experiments** are conducted for **each task**. For fairness, we directly compared with recently published works, adopting their experimental setups and data. Specifically:
>
> * For **restoration&super-resolution**
>
> Papers: UniFMIR (*Nature Methods* 2024), VCM (*WACV* 2025), and InverseSR (*MICCAI* 2023)
>
> Datasets: CARE (*Nature Methods* 2018), BioSR-CCPs/ERs/Microtubules/F-actin (*Nature Methods* 2021), HBA (*JNNP* 2003).
>
> * For **registration&fusion**
>
> Papers: Transmorph (*MIA* 2022), ConvexAdam (*TMI* 2024), and BSAFusion (*AAAI* 2025)
>
> Datasets: OASIS (*MICCAI2021*), IXI (*MIA* 2022), VIFB-CT&MRI/SPECT&MRI/PET&MRI (*TMM* 2023).
>
> In total, the imaging modalities directly covered in our experiments include Light Sheet Microscopy (LSM), Multimodal Structured Illumination Microscopy (M-SIM), CT, SPECT, PET, and MRI, with an even wider variety of imaging content. Objectively, we have done our best to ensure our experimental volume is substantial within the limited space of a NeurIPS submission. **Other reviewers have also acknowledged the breadth of our experiments** (**S4FJ**: *"Experimental efforts are comprehensive, with evaluation across many unique tasks, datasets, and modalities."*; **eeYn**: *"The effectiveness of the proposed method is validated through comprehensive performance comparisons with a variety of existing models."*). If you could point out a specific data type or study you are particularly interested in, we would be glad to include it for validation in a future version.
> ***
> **Q3:**
> > For the image fusion task, only three metrics are used to compare different methods, which is too small. Current image fusion studies typically utilize much more metrics.
>
> **A3:**
> We would first like to clarify that the three metrics were calculated following the methodology of BSAFusion (*AAAI* 2025) and   MSGFusion (*TMM* 2023), and we chose to display three key indicators for formatting and clarity. Since you did not specify which particular metrics or literature you would like us to include, we are providing additional metric results from our earlier experiments here for your reference.
> | Task | Methods | Qabf ↑ | Qcv ↓ | Qssim ↑ | Qvif ↑ | Qs ↑ |
> | :--- | :--- | :--- | :--- | :--- | :--- | :--- |
> | MRI-CT | BSAFusion | 0.39 | 4155.1 | 1.38 | 0.28 | 0.74 |
> | MRI-CT | Orochi | **0.41** | **2351.6** | **1.39** | **0.30** | **0.76** |
> | MRI-SPECT | BSAFusion | **0.77** | 110.8 | 1.07 | 0.42 | 0.79 |
> | MRI-SPECT | Orochi | 0.75 | **97.9** | **1.41** | **0.44** | **0.80** |
> | MRI-PET | BSAFusion | 0.78 | **67.7** | 1.05 | 0.42 | 0.77 |
> | MRI-PET | Orochi | **0.80** | 120.1 | **1.44** | 0.42 | **0.78** |
> ***
> **Q4:**
> > The training strategies of the proposed methods and the baselines are different in many experiments, which may make the comparison unfair.
>
> **A4:**
> We respectfully disagree with this statement. First, we clearly list our source database, codebase, and baseline scores in Section A for full transparency. Second, for all experiments, our method uses a **unified testing approach**, not a changing one. This involves three fine-tuning strategies: fully fine-tuning, replacing the decoder, and replacing with a lightweight decoder. The detailed framework diagram and experimental setup are provided in Figure 7 and Section B. If possible, we would appreciate it if the reviewer could point out which sentence or section may have led to the misunderstanding of an unfair comparison, and we will be sure to avoid any ambiguity in future revisions.
> ***
> **Q5:**
> > For the image registration task, there are a large set of different setting which bring different kinds of difficulty. Such as, registration between 2D images or 3D volumes, registration between different modality pairs or within the same modality, registration of different body parts with significant or not deformation, etc.
>
> **A5:**
> Our model actually has both 2D and 3D versions, and we have already performed various registration tasks in our paper, including 2D/3D, atlas-to-patient, and patient-to-patient registration (Table 3 & 7, Figure 4 & 9). The data we use is inherently multimodal—in addition to the original MRI, it also includes its multi-labeled mask. Following the standard training procedure of works like Transmorph, we fine-tune on this task using two multi-labeled masks for supervised registration learning, while inference is performed directly on the two MRI images.
>
> As requested, we are supplementing our results with an additional atlas-to-patient registration test on the multi-organ, four-dimensional extended cardiac-torso (XCAT) phantom data **[6]**. We have followed the same experimental setup as Transmorph, and the baseline results are taken from their paper. The final results are as follows:
> ### XCAT-CT (16 organs)
> | Model | Dice ↑ | % of &#124;Jφ&#124; ≤ 0 ↑ | SSIM ↑ |
> | :--- | :--- | :--- | :--- |
> | w/o registration | 0.220 ± 0.242 | - | 0.576 ± 0.071 |
> | Affined | 0.330 ± 0.291 | - | 0.751 ± 0.018 |
> | VoxelMorph-1 | 0.532 ± 0.313 | 2.275 ± 1.283 | 0.899 ± 0.027 |
> | VoxelMorph-2 | 0.548 ± 0.317 | 1.696 ± 0.909 | 0.910 ± 0.027 |
> | CycleMorph | 0.528 ± 0.321 | 3.263 ± 1.188 | 0.909 ± 0.024 |
> | TransMorph | 0.604 ± 0.314 | 1.679 ± 0.772 | 0.918 ± 0.023 |
> | Orochi | **0.621 ± 0.355** | **1.398 ± 1.049** | **0.924 ± 0.025** |
> ***
> **Reference**
>
> **[1]** Weigert, Martin, et al. "Content-aware image restoration: pushing the limits of fluorescence microscopy." Nature methods 15.12 (2018): 1090-1097.
>
> **[2]** Qiao, Chang, et al. "Evaluation and development of deep neural networks for image super-resolution in optical microscopy." Nature methods 18.2 (2021): 194-202.
>
> **[3]** Fang, Linjing, et al. "Deep learning-based point-scanning super-resolution imaging." Nature methods 18.4 (2021): 406-416.
>
> **[4]** Ma, Chenxi, et al. "Pretraining a foundation model for generalizable fluorescence microscopy-based image restoration." Nature Methods 21.8 (2024): 1558-1567.
>
> **[5]** Stringer, Carsen, and Marius Pachitariu. "Cellpose3: one-click image restoration for improved cellular segmentation." Nature methods 22.3 (2025): 592-599.
>
> **[6]** Segars, W. Paul, et al. "4D XCAT phantom for multimodality imaging research." Medical physics 37.9 (2010): 4902-4915.

---

> > ### Author Response · Authors · 2025-08-01
> >
> > A small arrow typo is in the header of Table XCAT-CT (16 organs)， the correct arrow should be as follows:
> >
> > ### XCAT-CT (16 organs)
> >
> > | Model | Dice ↑ | % of |Jϕ| ≤ 0 ↓ | SSIM ↑ |
> >
> > ...
> >
> > A smaller percentage of voxels with a non-positive Jacobian determinant (i.e., folded voxels) represent a more reasonable deformation grid produced by the model.

---

> > > ### Author Response · Authors · 2025-08-03
> > >
> > > ### Dear Reviewer,
> > >
> > > We are writing to follow up on our previous rebuttal.
> > >
> > > We would like to kindly mention that we have had a successful and thorough discussion with **Reviewer eeYn**, and we were able to resolve all of their concerns.
> > >
> > > Our rebuttal, which we prepared with considerable effort to address your insightful questions, is intended to facilitate a positive and comprehensive discussion with you. We believe the materials and new experiments directly address the points you raised.
> > >
> > > We look forward to your feedback and greatly appreciate the time you have dedicated to reviewing our work.
> > >
> > > Best regards,
> > >
> > > The Authors

---

> > ### Comment · Reviewer_Lxhu · 2025-08-07
> > **comments on response**
> >
> > I've carefully read the response by the authors, and my major concern about the insufficient validation has been clarified,especially with new results of more image fusion metrics and image registration tasks. Sorry for not noticing the results in the appendix. Therefore, I will raise my score.
> >
> > I’d like to clarify my comments about the low-level and high-level tasks. I did not mean that low-level tasks are useless, but there are some cases where the low level results are seldom used directly. For example, it is rare to fuse a CT image and a MRI image and present the results to a radiologist in clinical practice. In this case, the low-level metrics are not sufficient to evaluate the results. That’s why evaluation on high-level downstream tasks are more and more performed in image fusion studies. Anyway, this is acceptable since it is still a common experimental setup in image fusion studies and the proposed method has been evaluated from other perspectives.

---

> > > ### Author Response · Authors · 2025-08-07
> > >
> > > ### Dear Reviewer Lxhu
> > > We are very glad that we were able to reach an agreement in the end. We also appreciate you taking the time from your busy schedule to carefully review our rebuttal and for providing detailed examples that explained your initial concerns.
> > >
> > > We know that thoughtfully responding to a rebuttal is not easy, and we have learned new insights from your examples. This kind of discussion is undoubtedly positive and valuable.
> > >
> > > Again, thank you so much for your support!
> > >
> > > Best regards,
> > >
> > > The Authors

---

### Official Review · Reviewer_S4FJ · 2025-07-02

**Clarity:** 4
**Significance:** 3
**Originality:** 4
**Rating:** 6
**Confidence:** 4

**Summary:**

This paper presents a pretraining method and suite of models for a variety of biomedical image processing tasks such as restoration, super-resolution, and fusion. Orochi is trained on an extremely large (>14M) and diverse collection of medical images with an efficient, multi-scale representation learning approach tailored for downstream image processing tasks. When fine-tuned on downstream tasks, Orochi outperforms state-of-the-art foundation models and specialist models specifically tailored for individual modalities and tasks.

**Questions:**

1.	How does Orochi perform when fine-tuned on downstream (discriminative) tasks like classification? I understand that this does not fit under the umbrella of "low-level processing", but have the authors conducted such experiments? I would hypothesize it might perform well on dense tasks like segmentation given its multi-scale pretraining.

**Ethical Concerns:**

["NO or VERY MINOR ethics concerns only"]

**Final Justification:**

I thank the authors for their rebuttal and am encouraged to see strong results on downstream discriminative tasks. As indicated in my original review, I believe this is an excellent submission and will maintain my original rating of 6 (Strong Accept). I congratulate the authors on their great work and want to reiterate the importance of an open-source codebase and model for the research community.

**Limitations:**

Limitations are addressed in the appendix.

**Quality:**

4

**Strengths And Weaknesses:**

**Strengths**:
- The presentation quality is among the highest I have seen while reviewing for NeurIPS. Typesetting is used effectively to guide the reader’s attention and break up passages of text. Figures are dense, informative, and visually appealing
- The paper is both organized and written clearly. Prior work and important background are thoroughly explained.
- Experimental efforts are comprehensive, with evaluation across many unique tasks, datasets, and modalities. Moreover, performance is compared against state-of-the-art specialized baseline approaches
- Open-source release of code and pre-trained weights should make this a major contribution to the field

**Weaknesses**:
- I see no major weaknesses. However, I am curious to see how Orochi performs on downstream discriminative tasks like classification or even segmentation.

**Minor comments**:
- I originally misinterpreted the word “study”. In the context of medical imaging, this can often effectively mean “exam”; thus, “100 studies” seemed to be a very small amount. Context clues clarified this confusion quickly, but I would suggest potentially changing the wording here. There is probably a better way to clarify that this is an aggregate of many existing datasets.
- Related to the above point, I would try to highlight the “size” (e.g., # images) of the composite pretraining dataset as early as possible to convey the scale and dispel any confusion
- Figure 1: While funny (to me, at least), I’m not sure “Crappifier” is appropriate
- L172: Change “down-sample” -> “Down-sampling”
- L201 is not a complete sentence: “Resulting in more than 30…” perhaps simply remove the period and write “resulting”
- Table 5: The title is confusing, “Pre-train Strategies V.S Performance”. Should this read “Vs.” as in “versus”?
- Figure 5: Visually appealing, but difficult to read. Perhaps a different font/color choice is needed or the figure should be enlarged.

---

> ### Author Rebuttal · Authors · 2025-07-27
>
> We would like to extend our sincerest gratitude to the reviewer for the insightful feedback and high praise for our manuscript. We are truly honoured that our work is considered **outstanding among many excellent submissions.** A great deal of effort was dedicated to structuring the narrative, creating the figures, and designing the experiments, all with the goal of clearly and efficiently communicating the focus and contributions of our work within the limited space. It is incredibly **encouraging to have our efforts recognized, regardless of the final outcome.**
>
> We have carefully considered all the comments and provide the following point-by-point responses:
> ***
> **Q1:**
> > I originally misinterpreted the word “study”. In the context of medical imaging, this can often effectively mean “exam”; thus, “100 studies” seemed to be a very small amount. Context clues clarified this confusion quickly, but I would suggest potentially changing the wording here. There is probably a better way to clarify that this is an aggregate of many existing datasets.
>
> **A1:**
> Thank you for this valuable suggestion. We used the term “study” because our primary data source, the Image Data Resource (IDR) **[1]**, uses this term to describe the original user-uploaded source data. Currently, the resource contains 138 studies, which comprise 14,042,467 images and a total of 414TB of data. To prevent ambiguity, we will avoid using the term “study” when describing other data sources in our future versions of the manuscript.
> ***
> **Q2:**
> > Related to the above point, I would try to highlight the “size” (e.g., # images) of the composite pretraining dataset as early as possible to convey the scale and dispel any confusion.
>
> **A2:**
> This is an excellent point. We agree and will be sure to emphasize the scale of the dataset in future versions of the manuscript.
> ***
> **Q3:**
> > Figure 1: While funny (to me, at least), I’m not sure “Crappifier” is appropriate.
>
> **A3:**
> Thank you for your feedback. The term “Crappifier” was adopted from the *Nature Methods* publication, ***"Deep learning-based point-scanning super-resolution imaging"*** **[2]** which systematically investigates the suitability of various synthetic noise and artifacts for real-world microscopy image restoration and super-resolution. This paper was a significant reference for us when integrating self-supervised methods for these tasks. However, your point is well-taken; given that our work extends beyond image restoration and super-resolution, a more neutral term such as “Converter” would indeed be more appropriate. We will make this change in our revision.
> ***
> **Q4:**
> > L172: Change “down-sample” -> “Down-sampling”
> > L201 is not a complete sentence: “Resulting in more than 30…” Perhaps simply remove the period and write “resulting”
>
> **A4:**
> Thank you for your meticulous reading. We will correct these in the next version.
> ***
> **Q5:**
> > Table 5: The title is confusing, “Pre-train Strategies V.S Performance”. Should this read “Vs.” as in “versus”?
>
> **A5:**
> Thank you for pointing this out. Your understanding is correct, and we will revise the title accordingly.
> ***
> **Q6:**
> > Figure 5: Visually appealing, but difficult to read. Perhaps a different font/colour choice is needed, or the figure should be enlarged.
>
> **A6:**
> We appreciate this feedback. The page limit constrained the current size of the figure for the ablation study. We will revise it for better readability in a future version of the manuscript where more space is available.
> ***
> **Q7:**
> >I see no major weaknesses. However, I am curious to see how Orochi performs on downstream discriminative tasks like classification or even segmentation. (Weakness #1)
>
> >How does Orochi perform when fine-tuned on downstream (discriminative) tasks like classification? I understand that this does not fit under the umbrella of "low-level processing", but have the authors conducted such experiments? I would hypothesize it might perform well on dense tasks like segmentation, given its multi-scale pertaining. (Question #1)
>
> **A7:**
> We conducted new tests on the model's performance on BTCV **[3]**, REFUGE **[4]**, and CadVidSet **[5]** segmentation datasets. We strictly adhered to the training hyperparameter settings of MedSAM2 **[6]** and carried out a complete fine-tuning following its GitHub repository. Baseline results are from the original paper.
> ### Dice Score
> | Model | REFUGE-2D | CadVidSet-2D | BTCV-Aorta-3D | BTCV-Liver-3D |
> | :--- | :--- | :--- | :--- | :--- |
> | TransUNet | 0.363 | 0.316 | **0.920** | 0.969 |
> | Swin-UNet | 0.289 | 0.226 | 0.892 | **0.975** |
> | nnUNet | 0.349 | 0.418 | 0.877 | 0.948 |
> | SAM2 | 0.558 | 0.539 | 0.835 | 0.861 |
> | MedSAM2 | 0.799 | **0.861** | 0.896 | 0.865 |
> | Orochi | **0.811** | 0.857 | 0.911 | 0.950 |
>
> Also, we conducted classification tasks on TissueMNIST (2D) and OrganMNIST (3D) datasets. All experimental setups are from the MedMNIST series papers **[7] [8] [9]**.
>
> | Methods (TissueMNIST-2D-224^2) | AUC | ACC | Methods (OrganMNIST-3D-28^3) | AUC | ACC |
> | :--- | :--- | :--- | :--- | :--- | :--- |
> | ViT-Base | 0.948 | 0.729 | ResNet-18+3D | **0.996** | **0.907** |
> | MAE-Base | 0.945 | 0.719 | ResNet-50+3D | 0.994 | 0.883 |
> | CLIP-Base | 0.922 | 0.662 | ResNet-50+2.5D | 0.974 | 0.769 |
> | EVAv2-Base | 0.940 | 0.703 | Auto-sklearn | 0.977 | 0.814 |
> | DINOv2-Base | 0.944 | 0.715 | AutoKeras | 0.979 | 0.804 |
> | Orochi | **0.950** | **0.733** | Orochi | 0.995 | 0.891 |
>
> We demonstrated the capability of Orochi to adapt to high-level tasks; however, we aim to avoid overstating its application. Our objective remains focused on the development of a unified low-level processor, as stated since the abstract of our paper.
> ***
> **Reference**
>
> **[1]** Williams, Eleanor, et al. "Image Data Resource: a bioimage data integration and publication platform." Nature methods 14.8 (2017): 775-781.
>
> **[2]** Fang, Linjing, et al. "Deep learning-based point-scanning super-resolution imaging." Nature methods 18.4 (2021): 406-416.
>
> **[3]** Landman BA, Xu Z, Igelsias JE, Styner M, Langerak TR, and Klein A, "MICCAI multi-atlas labeling beyond the cranial vault workshop and challenge," (2015)
>
> **[4]** Orlando, José Ignacio, et al. "Refuge challenge: A unified framework for evaluating automated methods for glaucoma assessment from fundus photographs." Medical image analysis 59 (2020): 101570.
>
> **[5]** Wang, Lu, et al. "Coronary artery segmentation in angiographic videos utilizing spatial-temporal information." BMC medical imaging 20.1 (2020): 110.
>
> **[6]** Zhu, Jiayuan, et al. "Medical sam 2: Segment medical images as video via segment anything model 2." arXiv preprint arXiv:2408.00874 (2024).
>
> **[7]** Yang, Jiancheng, et al. "Medmnist v2-a large-scale lightweight benchmark for 2d and 3d biomedical image classification." Scientific Data 10.1 (2023): 41.
>
> **[8]** Yang, Jiancheng, Rui Shi, and Bingbing Ni. "Medmnist classification decathlon: A lightweight automl benchmark for medical image analysis." 2021 IEEE 18th International Symposium on Biomedical Imaging (ISBI). IEEE, 2021.
>
> **[9]** Doerrich, Sebastian, et al. "Rethinking model prototyping through the MedMNIST+ dataset collection." Scientific reports 15.1 (2025): 7669.

---

### Note · Authors · 2025-08-11

### Dear SAC, AC and Reviewers,

We would like to summarize our submission briefly.

All of the reviewers who have provided a **final rating** endorsed our paper with support.

**A. Quantitative scores** (Quality, Clarity, Significance, Originality):
- `S4FJ`: 4 (excellent), 4, 3, 4
- `Lxhu` and `eeYn`: 3 (good), 3, 3, 3.

**B. Qualitative feedbacks**
- **Presentation**  - ***"Highest I have seen while reviewing for NeurIPS."*** said by `S4FJ`. Besides, `Lxhu` said
> "The paper is well written and easy to follow."

- **Data** - ***"Extremely large and diverse collection."*** said by `S4FJ`.
`Lxhu` and `eeYn` also acknowledged:
> "A large-scale dataset collected."

- **Methods** - ***"Motivation is reasonable and valuable. Pre-training strategy is suitable."*** said by `Lxhu`.
`eeYn` and `S4FJ` also summarized:
> "Orochi is a valuable contribution, especially in its focus on generality and multi-task capability."

- **Experiments** - ***"Comprehensive, with many unique tasks, datasets, and modalities."*** said by `S4FJ`.
`eeYn` supports this by:
> "Validated through comprehensive performance comparisons with a variety of existing models."

`Z17s` has not yet provided a final rating. Nevertheless, we appreciate receiving recognition for our work as a demonstration of the "scaling law". We have provided detailed clarifications and additional experiments in both our initial rebuttal and follow-up discussion to address their uncertainty.

**C. Our Summary**

It is challenging for biologists to find appropriate tools from a fragmented "toolbox" of specialized image processing models. In response, Orochi aim to develop a "Swiss Army knife"-style solution, showing the potential of a unified, multitask-capable model for the community. We introduce a scalable data collection method, which was employed to compile 2D and 3D samples from raw data across more than 100 studies from IDR and multiple Atlas datasets. This extensive dataset serves as the foundation for training our efficient Mamba-based framework, enhanced by a novel Task-related Joint-embedding Pre-training strategy inspired by integrating common degradation patterns in biomedical imaging. We compared a comprehensive set of recently published models from top-tier conferences and journals. A diverse range of tasks was evaluated (e.g., Registration, Fusion, Restoration, and Super-Resolution). Our approach demonstrates superior effectiveness and efficiency compared to these models.

### The Authors

---

### Decision · Program_Chairs · 2025-09-17

**Decision:**

Accept (spotlight)

**Comment:**

This paper present a new foundation model for biomedical image processing and evaluates it on a diverse range of tasks. Reviewers praised the breadth of the experimental results and clarity of the presentation while raising concerns about the feasibility of transfer from a single model to such a diverse array of tasks.

While the reviews are varied in score, comprising every number in the range 3-6, the borderline positive reviewer gave high praise to the work (using the word "outstanding") and noted that their low confidence prevented them from giving a higher score. Meanwhile, the negative reviewer's final justification raised concerns about whether the model could truly be transferred to diverse tasks that seem to be addressed by the work's empirical results and did not respond to the authors' (or the AC's) pushback. Therefore I believe the paper warrants acceptance.